# Carbon and nitrogen pools in thermokarst-affected permafrost landscapes in Arctic Siberia

Matthias Fuchs[1,2], Guido Grosse[1,2], Jens Strauss[1], Frank Günther[1], Mikhail Grigoriev[3], Georgy M. Maximov[3], Gustaf Hugelius[4]

[1]Alfred Wegener Institute Helmholtz Centre for Polar and Marine Research, 14473 Potsdam, Germany
[2]Institute of Earth and Environmental Sciences, University of Potsdam, 14476 Potsdam, Germany
[3]Melnikov Permafrost Institute, 677010 Yakutsk, Russia
[4]Department of Physical Geography, Stockholm University, 106 91 Stockholm, Sweden

*Correspondence to*: Matthias Fuchs (matthias.fuchs@awi.de)

**Abstract.** Ice rich Yedoma-dominated landscapes store considerable amounts of organic carbon (C) and nitrogen (N) and are vulnerable to degradation under climate warming. We investigate the C and N pools in two thermokarst-affected Yedoma landscapes – on Sobo-Sise Island and on Bykovsky Peninsula in the North of East Siberia. Soil cores up to three meters depth were collected along geomorphic gradients and analysed for organic C and N contents. A high vertical sampling density in the profiles allowed the calculation of C and N stocks for short soil column intervals and enhanced understanding of within-core parameter variability. Profile-level C and N stocks were scaled to the landscape level based on landform classifications from five-meter resolution, multispectral RapidEye satellite imagery. Mean landscape C and N storage in the first meter of soil for Sobo-Sise Island is estimated to be 20.2 kg C m$^{-2}$ and 1.8 kg N m$^{-2}$ and for Bykovsky Peninsula 25.9 kg C m$^{-2}$ and 2.2 kg N m$^{-2}$. Radiocarbon dating demonstrates the Holocene age of thermokarst basin deposits but also suggests the presence of thick Holocene aged cover layers which can reach up to two meters on top of intact Yedoma landforms. Reconstructed sedimentation rates of 0.10 mm yr$^{-1}$ - 0.57 mm yr$^{-1}$ suggest sustained mineral soil accumulation across all investigated landforms. Both Yedoma and thermokarst landforms are characterized by limited accumulation of organic soil layers (peat).

We further estimate that an active layer deepening by about 100 cm will increase organic C availability in a seasonally thawed state in the two study areas by ~5.8 Tg (13.2 kg C m$^{-2}$). Our study demonstrates the importance of increasing the number of C and N storage inventories in ice-rich Yedoma and thermokarst environments in order to account for high variability of permafrost and thermokarst environments in pan-permafrost soil C and N pool estimates.

## 1 Introduction

Yedoma regions store large amounts of soil organic carbon (SOC) and are highly vulnerable to permafrost thaw under climate warming (Zimov et al., 2006, Strauss et al., 2013). Organic C, freeze-locked for millennia in these permafrost deposits, may become available for increased microbial activity and release in the form of the greenhouse gases $CO_2$ and $CH_4$ after thaw (Gruber et al., 2004; Tarnocai, 2006; Dutta et al., 2006, Schuur et al., 2008; 2015). Carbon-rich Yedoma regions therefore

have the potential to contribute significantly to greenhouse gas emissions in a rapidly warming Arctic (Schneider von Deimling et al., 2015).

The term Yedoma describes late-Pleistocene, ice-rich, silty deposits covering vast areas in the northern permafrost regions that were not glaciated during the last ice age (Schirrmeister et al., 2011a; 2013). These deposits can reach a thickness of up to 50 m, store large amounts of C, and are still present today in an area of approximately 625,000 km$^2$ in the Siberian and North American permafrost region (Strauss et al., 2017). Since deglacial warming, large areas of the former Yedoma surfaces, expanding across several million km$^2$ of North and East Siberia, Alaska, Northwest Canada, and the adjacent shelf regions, were eroded by thermokarst processes. These processes affected the landscape by permafrost thawing and ice wedge melting leading to surface lowering, lake formation, and lake drainage. Resulting landscape features include thermokarst mounds (also called baydzherakhs), thermo-erosional gullies, thermokarst lakes, and thermokarst depressions (drained thaw lake basins or alas) which now are dominant features in the Yedoma terrain (Jorgenson and Shur, 2007; Veremeeva and Gubin, 2009; Kokelj and Jorgenson, 2013, Morgenstern et al., 2013, Ulrich et al., 2014).

Ice rich Yedoma and thermokarst deposits are prone to thaw after disturbances that change the surface thermal regime (e.g. from changing air temperatures, fire, or anthropogenic disturbances). Since Yedoma as well as thermokarst deposits contain large, potentially vulnerable C pools in permafrost regions (Zimov et al., 2006; Strauss et al., 2013; Walter Anthony et al., 2014) they are of global importance for SOC accumulation, degradation, and release. Whereas degrading Yedoma surfaces and thermokarst features lead to thaw-induced remobilization of organic C, the drainage of thermokarst lakes and the following permafrost aggradation lead to SOC accumulation and stabilization (Hinkel et al., 2003; Jorgenson and Shur, 2007; Jones et al., 2012; Grosse et al., 2013). Olefeldt et al. (2016) estimated that landscapes potentially vulnerable to thermokarst contain up to 50% of the total SOC stored in the permafrost region. In the Yedoma region, 60% of the C is stored in drained and refrozen thermokarst basins (Strauss et al., 2013). However, these estimates are based on few data points from the Yedoma region and it remains important to enhance our understanding of the C storage, distribution, and vulnerability in these environments with detailed landscape-scale assessments.

Several permafrost C pool estimates have been carried out from different areas within the Arctic region (e.g. Michaelson et al., 1996; Kuhry et al., 2002; Ping et al., 2008; Tarnocai et al., 2009; Horwath Burnham and Sletten, 2010; Hugelius et al., 2010, 2011; Fuchs et al., 2015; Siewert et al., 2015; 2016; Palmtag et al., 2016). A synthesis for the circum-Arctic by Hugelius et al. (2014) merged a large number of datasets into the Northern Circum-polar Soil C Database (NCSCD) resulting in an estimate of 1035 ± 150 Pg of organic C for 0-3 m for soils in the permafrost region. Several additional estimates for SOC storage in the Yedoma region exist, some of which are taking into account slightly different components of the Yedoma region. Zimov et al. (2006) calculated 450 Pg C for the C pool for the entire Yedoma region (including both Yedoma and thermokarst deposits). Walter Anthony et al. (2014) estimated the total Holocene and Pleistocene soil C pools of the Yedoma region with 429 ± 101 Pg C, while Hugelius et al. (2014) calculated 181 ± 54 Pg C for all deposits in the Yedoma region below three meter depth and Strauss et al. (2013) calculated 211 + 160/-153 Pg C for the entire Yedoma deposits including the top three meters. Despite the variation in these estimates they all suggest a very large C pool of several hundred

Pg for this region and confirm that these ice-rich deep deposits are a globally important C pool in the northern circumpolar permafrost region. Detailed local studies for particular parts of the Yedoma region are scarce so far but suggest significant landscape-scale and inter-regional variation in SOC stocks that warrant further local studies and regional syntheses (Schirrmeister et al., 2011b, 2011c; Strauss et al., 2012; Siewert et al., 2015, 2016; Shmelev et al., 2017, Webb et al. 2017).

However, the Yedoma and thermokarst landscapes not only host important C pools but potentially store a significant amount of N. Even though there are several permafrost soil C studies, only very few report N stocks (Ping et al., 2011; Harden et al., 2012; Michaelson et al., 2013; Zubrzycki et al., 2013; Obu et al., 2015, Palmtag et al., 2015). In tundra environments, N often is the limiting factor for plant growth (Shaver et al., 1986; Chapin et al., 1995; Mack et al., 2004; Beermann et al., 2015). As a result, thawing permafrost not only has the potential to release large amounts of organic C but also to increase the

availability for N which may increase primary production and partly offset increased greenhouse gas emissions from permafrost soils (e.g. Keuper et al., 2012, 2017; Natali et al., 2012, Salmon et al., 2016).

           A potential indicator for the potential C loss upon thaw and the decomposability of C stored in permafrost soils is the carbon-nitrogen (CN) ratio (Schädel et al., 2014). Generally, a higher CN ratio suggests less degraded organic C, while a lower CN ratio points towards already partly degraded C before incorporation into permafrost (Weiss et al., 2016). Even though the

15 CN ratio cannot be taken as a single variable to estimate the recalcitrance of the soil organic matter, it provides a first indication of the potential degradability of C stored in permafrost soils (Kuhry and Vitt, 1996; Hugelius and Kuhry, 2009; Hugelius et al., 2012). Generally, ice-rich landscapes are underrepresented in pan-Arctic permafrost C pool inventories and the variability of these environments is poorly accounted for. Especially the deep C and N stocks below one meter depth remain associated with high uncertainties due to low sample numbers. In addition, thermokarst areas are temporally dynamic and heterogeneous

environments whose characteristics are difficult to generalize and characterize. A wide range of local case studies is needed to capture spatial variability and to improve C pool estimations. This study presents new data on C and N storage in two Yedoma regions in the North of East Siberia and analyzes the variability of C and N contents along landscape gradients extending from Yedoma uplands into adjacent drained thaw lake basins (DTLB), investigating whether C and N storage is significantly higher in DTLBs than in Yedoma soils as proposed in regional studies by Strauss et al. (2013), or Walter Anthony et al. (2014).

In this study, sediment cores up to 3 meter depth from typical thermokarst-affected Yedoma regions were analyzed 1) to quantify the organic C and N variability with depth and along geomorphologic gradients in thermokarst-affected terrain; 2) to understand depositional chronologies as well as C accumulation rates; and 3) in combination with a satellite image based terrain classification to quantify landscape-scale SOC and N inventories for the study areas Bykovsky Peninsula and Sobo-Sise Island.

## 2 Material and methods

### 2.1 Study area

The two study areas are situated in the North of East Siberia in the Lena River Delta region (Fig. 1). The first study site, Sobo-Sise Island (72°29' N, 128°16' E), is a Yedoma remnant within the eastern Lena Delta. These remnants are commonly referred to as the third geomorphological river delta terrace (Schwamborn et al., 2002). According to Morgenstern et al. (2011), Sobo-Sise Island accounts for almost 16% of the entire Yedoma coverage in the Lena Delta. It is characterized by Yedoma uplands but also features permafrost degradation landforms such as thermokarst lakes, drained thaw lake basin, and thermo-erosional gullies. Sobo-Sise is located between two of the main channels of the Lena Delta (Sardakhskaya and Bykovskaya channels), elongated in east-west direction (45 km) and has an area of 336 km$^2$. Very few studies have been conducted on the depositional characteristics of Sobo-Sise. For example, Grigoriev (1993) dated a mammoth bone from a Yedoma cliff on Sobo-Sise and Grigoriev (2007) investigated the shore erosion on Sobo-Sise's coastline. Morgenstern et al. (2011) as well as Nitze and Grosse (2016) included Sobo-Sise in regional remote sensing studies aiming at the quantification of thermokarst lakes and basins and surface landscape changes, respectively.

The second study site, Bykovsky Peninsula (71°51' N, 129°19' E), is similar to Sobo-Sise and also dominated by Yedoma uplands that are intersected by thermokarst lakes, drained thermokarst lake basins and thermo-erosional gullies. The Bykovsky Peninsula is located to the southeast of the Lena River Delta at the Laptev Sea coast and represents an erosional remnant of a Late Pleistocene Yedoma accumulation plain blanketing the foreland of low mountain ridges (Grosse et al., 2007). The peninsula is a narrow land tongue with a width between 1 – 4 kilometers and has an area of 167 km$^2$. In contrast to Sobo-Sise Island in the Lena Delta, the Bykovsky Peninsula is strongly affected by rapid coastal erosion from all sides and by coastal inundation, manifested in several thermokarst lagoons. Studies on the Bykovsky Peninsula focused mostly on paleoenvironmental reconstructions using Late Quaternary deposits at the exposure Mamontovy Khayata (Kunitsky, 1989; Slagoda 1991, 1993; Griogriev, 1993; Siegert et al., 2002; Meyer et al., 2002; Schirrmeister et al., 2002a). Kholodov et al. (2006) described organic matter characteristics in Alas and Yedoma deposits, Griogriev et al. (1996) and Lantuit et al. (2011) determined coastal erosion rates on the coasts of Bykovsky Peninsula, and Grosse et al. (2005) mapped the periglacial geomorphology based on remote sensing data.

The climate of Sobo-Sise Island and Bykovksy Peninsula is continental, despite their proximity to the coast. Both sites are dominated by polar tundra climate (Peel et al., 2007). The mean annual air temperature (MAAT; period 1998-2011) on Samoylov Island (the closest weather station to Sobo-Sise, around 50 km to the west) is -12.5°C with an approximate mean annual precipitation of 180-200 mm (including mean summer rainfall and mean snow water equivalent, period 1998-2011) (Boike et al., 2013). On Bykovsky, the climate is similar with a MAAT of -12.9°C (period 1933-2013) at the closest weather station in Tiksi (Günther et al., 2015), but a higher mean annual precipitation with around 240-260 mm (Grigoriev, 1993). The higher precipitation can be explained by the influence of the Laptev Sea and the mountain ridge nearby in the hinterland (Boike

et al., 2013). In both study sites, permafrost is continuous and is assumed to have a thickness of about 500-650 m (Griogriev, 1993). Boike et al. (2013) reported a mean annual permafrost temperature at Samoylov Island to be -8.6°C in 10.7 m depth.

In both study sites, the tundra vegetation is sparse and is limited by the short growing season (June to September). The vegetation composition is dominated by sedges, grasses, mosses, lichen and sporadic dwarf shrubs (*salix* species). According to Schneider et al. (2009) most of Sobo-Sise Island is classified as moist grass- and moss-dominated tundra and wet sedge- and moss-dominated tundra. In the Circum Arctic Vegetation Map (Walker et al., 2005), large parts of the Lena River Delta including Sobo-Sise Island are classified as sedge, moss, dwarf shrub wetland and Bykovksy Peninsula is classified as non-tussock sedge, dwarf-shrub, moss tundra.

Following the US Soil Taxonomy (Soil Survey Staff, 2014) the soil types in both study sites belong to the Gelisol order with its suborders Turbels and Orthels. Only two soil profiles sampled in thermokarst deposits in this study contain thicker organic layers and can be considered as Histels. Permafrost occurs in almost all sampled sites within the first meter. Active layer thickness ranges from a minimum of 18 cm in thermokarst deposits to a maximum of 84 cm in sandy fluvial deposits with a mean of all sampled sites of 40 cm (median 41 cm). Only in two sites in non-vegetated, sandy, temporally flooded Lena River floodplain deposits of Sobo-Sise, permafrost was not present near the surface and these soils can be classified as Entisols.

## 2.2 Field work

Soil samples were collected in August 2014 along four transects of 500 m (BYK14-T3) and 600 m (BYK14-T2, SOB14-T1, SOB14-T2) length. We chose transects to cover key geomorphologic gradients of the thermokarst-affected landscapes starting on Yedoma uplands and passing through different stages of Yedoma degradation (including Yedoma slopes) and stages of the thermokarst lake cycle. Similar transect-based approaches were used in other regions of Siberia (Siewert et al., 2015; Palmtag et al., 2015), Canada (Hugelius et al., 2010) and Alaska (Jorgenson, 2000; Kanevskiy et al., 2014). We determined the starting points as well as the direction of each transect. To include randomness, the sampling points along the transects were then chosen at equidistant intervals in alignment with the scale of landscape features at a site. On the first transect (SOB14-T1) a 150 m and on the second transect (SOB14-T2) a 100 m distance between sampling points was chosen. The first transect on Sobo-Sise extended from a Yedoma upland into the adjacent DTLB. The second transect extended from a Yedoma upland, crossing a DTLB and ending in fluvial deposits of the floodplain. Additional non-random sample points were collected on a baydzherakh (erosional remnants of polygon centers forming thermokarst hills) and two DTLBs. No cores were taken from extant thermokarst lakes or thermo-erosional gullies. On Bykovsky Peninsula, transects were chosen with the same approach with a 150 m distance on the first transect (BYK14-T2) and 100 m on the second transect (BYK14-T3) between the sampling points. BYK14-T2 runs from one DTLB over a Yedoma remnant covered with baydzherakhs into another adjacent DTLB. BYK14-T3 again was similar to the Sobo-Sise transects, running from top of a Yedoma upland into an adjacent DTLB.

Along the transects, soil pits were excavated down to the bottom of the active layer. A soil profile description was made and fixed-volume samples were collected with a metal cylinder of known volume. After sampling the active layer,

permafrost cores and samples were collected with a SIPRE (Snow, Ice and Permafrost Research Establishment) auger barrel drill (Jon´s Machine Shop, Fairbanks, USA) with a diameter of 7.62 cm (3"). Total sampling depths ranged from 45 cm to 318 cm. As a result, at 23 sites the whole soil profile was sampled and characterized including active, transient and permafrost layers and at five additional sites only the active layer was sampled. A permafrost core description was made and the cores were subsampled in the field in five to ten centimeter intervals depending on facies horizons. The visual core description included cryostratigraphy according to French and Shur (2010) as well as a description of the lithology and plant macrofossils.

## 2.3 Laboratory analysis

In total 455 samples were processed for total carbon (TC), total nitrogen (TN) and total organic carbon (TOC). Samples were freeze-dried, homogenized by grinding prior to measurement of 5-8 mg of the samples with a Vario EL III Elemental Analyzer for TC and TN. Both, %C and N were measured in one run. We measured two replicates of each sample where we accept a <5% deviation for the two measurements. The sensitivity of the Elemental Analyzer is < 0.1%. Afterwards, samples were measured for TOC (15-100 mg, depending on the TC content) with a Vario Max C Elemental Analyzer. Again, we allow a deviation of <5% for the double measurements. Total inorganic carbon (TIC) is then calculated as the difference between TC and TOC. In addition, the CN ratio was calculated as quotient between TOC and TN contents.

Based on Hugelius et al. (2010), the (SOC) storage was calculated for each sample using Eq. (1):

$$SOC\ [kg/m^2]\ =\ TOC\ [\%]\ x\ BD\ [g/cm^3]\ x\ (1-CF)\ x\ length\ [cm]\ x\ 10 \tag{1}$$

Here, TOC is the total organic C content derived from the elemental analysis, BD is the dry bulk density of the sample, CF is the coarse fragment fraction ($\varnothing > 2$ mm) (which was zero because of the absence of coarse fragments in the sampled soils), and length is the actual sample length. The sample-specific SOC contents were added up to the reference depths of 0 – 30 cm, 0 – 100 cm, 0 – 200 cm. If the cores were not recovered completely, missing core intervals, or respectively missing SOC contents, were interpolated between adjacent samples or from samples with the same characteristics following the field notes. The ice content in a sample is reflected in the dry bulk density and therefore included in the calculations. The SOC contents for the different reference depths however do not include ice wedge volumes. Ice wedge volume was included in a later step when scaling site-specific SOC stocks to the landscape level. Likewise the N storage was calculated for individual samples and for the different reference depths.

In addition, 25 sub-samples from various depths of seven different sampling sites were chosen for radiocarbon dating of organic macro fossils. All samples were wet-sieved with a 2µm sieve and plant macro remains (mostly moss leaves or sedge stems) were handpicked under a microscope. In a few cases with insufficient macro remains, bulk samples were selected. Samples were then submitted to the Radiocarbon Laboratory in Poznan, Poland, where the samples were analyzed and dated with the accelerated mass spectrometer (AMS) dating method (Goslar et al., 2004). The obtained radiocarbon ages were eventually calibrated with the Calib 7.1 software to calibrated radiocarbon years before present (cal. a BP) (Stuiver and Reimer, 1993; Stuiver et al., 2017).

## 2.4 Landform classification and upscaling C and N pools

All geospatial analyses were performed in the ESRI ArcGIS 10.1 and ENVI 5.3 software. For both Sobo-Sise Island and Bykovsky Peninsula, multispectral RapidEye Images (pixel resolution 5 m) in combination with high resolution digital elevation models (DEM) were used to classify the landscape into the dominant landscape features. For Sobo-Sise Island two scenes from the same date (27th July 2014) were orthorectified and mosaicked together to cover the entire island. To increase the data basis, the RapidEye mosaic from the 27th July 2014 was stacked with another RapidEye scene (acquisition: 30th June 2014) which covers the entire island. The same scenes (27th July and 30th June 2014 images) were also used for Bykovsky Peninsula, stacked together with an additional scene acquired at 9th Sept. 2014 since the first scene does not cover the entire peninsula. This procedure assures that additional landform variability from the phenological stages of the tundra vegetation at both study sites are captured by at least two RapidEye scenes entirely.

A maximum likelihood supervised classification relying on training areas was used to classify the landscape into the predominant landscape features. Identification of training areas also has been facilitated through near-simultaneous acquisition of RapidEye imagery to our field work. Training areas were chosen based on field notes and field knowledge. For Sobo-Sise Island the different RapidEye images were stacked with a DEM (pixel resolution 2 m) that had been derived from photogrammetric processing of three overlapping GeoEye stereo image pairs (acquisition dates range from 27th July to 15th August 2014). For Bykovsky Peninsula, a DEM was derived from overlapping WorldView-1 and WorldView-2 stereo image pairs (acquisition period: 25th to 29th Aug. 2015) and again combined with the 15 band deep multispectral RapidEye image stack to run the maximum likelihood classification. Adding the DEM allowed enhanced classification of Yedoma uplands, since plant communities on Yedoma uplands cannot be entirely distinguished from those in DTLBs based only on spectral signatures. This advantage was already demonstrated by Grosse et al. (2006) and Siewert et al. (2016) who showed that by including a DEM non-degraded Yedoma uplands and partly degraded Yedoma uplands could be better discriminated compared to image classification only.

The two landform classification for Sobo-Sise and Bykovsky Peninsula initially included the main geomorphological units *Yedoma upland*, *partly degraded Yedoma* (Yedoma slope), and *DTLBs* of different generations. However, due to the small amount of collected sampling sites in DTLBs of different generations, the landform classes of DTLBs were merged to a single class 'Thermokarst' for the upscaling. The final landform classification used for upscaling included the following classes: *Yedoma uplands*, *Degraded Yedoma / Yedoma slope*, *Thermokarst*, and *Lakes*. The areas of lakes were excluded from upscaling since this study focuses on terrestrial soil C storage and no lake cores were collected.

The landform classification accuracy was based on field-based ground truthing points complementary with data points extracted from high resolution imagery. The high resolution imagery include a GeoEye (Sobo-Sise Island), WorldView2 (Bykovsky Peninsula) and aerial photography to assess the correctness of classification and overlap with the sampled field sites. For each study area, 300 randomly selected points and an additional 20 points for each landform class were manually classified and then compared with the landform classification based on the RapidEye satellite imagery.

The total SOC and N storage for Sobo-Sise and Bykovsky Peninsula was based on mean C stocks of the collected sampling sites for the reference depths 0-30 cm, 0-100 cm, and 0-200 cm of each class. The mean stocks were upscaled based on the areal extent of the corresponding landform class. This approach allows a first estimate of the potential C and N storage in the study areas for the first two meters of soil. Confidence intervals for the mean SOC and N landscape stocks were calculated according to Hugelius (2012). However, these confidence intervals do not include uncertainties evolving from the landform classification based upscaling.

To not overestimate the C and N stocks in the upscaling, we accounted for the ice wedge volume in the landscape carbon and nitrogen calculations. Ice wedge contents for thermokarst deposits were adopted from the study of Ulrich et al. (2014) as a mean value derived from the maximum ice wedge contents from their three Northeast Siberian study areas Ebe-Basyn-Sise, Cape Mamontov Klyk and Buor Khaya Peninsula. This resulted in a mean ice wedge volume of 9% $\pm$ 3% (standard deviation) for thermokarst deposits which is similar to what Kanevskiy et al. (2013) found for drained lake basins in Arctic Alaska with 8%. For the estimation of Yedoma upland ice wedge volume, the GIS based approach from Günther et al. (2015) was applied for Sobo-Sise. We ortho-rectified the GeoEye image using our high resolution DEM, in order to ensure consistent mapping in an image free of geometrical distortions. We mapped more than 1500 baydzerakhs that appeared in clusters on slopes around thermokarst lakes and along thermo-erosional valleys and river banks to determine the spatial dimensions of ice wedge polygons (Voronoi diagram). The largest possible circle within each polygon served as a proxy for the sediment fraction of the polygon and was put in relation to the remaining size of the polygon, representing the ice wedge fraction. This resulted in a mean ice wedge volume of 40% $\pm$ 8% (standard deviation). For Yedoma uplands on Bykovsky Peninsula an ice wedge volume of 44% was applied, which is the mean macro ground ice content on nearby Muostakh Island (Günther et al., 2015). Both, Muostakh Island and Bykovsky Peninsula have formerly been connected with each other (Grigoriev, 1993) and are now separated by a 15 km wide sound. For the active layer, we assumed no ice wedge volume for the calculations. Mean active layer depths for the different classes were derived from the collected soil sites. Mean active layer depths for the sampled sites are presented in the supplementary material Table S1.

## 3 Results

### 3.1 Sedimentological results

Table 1 presents the laboratory results, indicating that samples from the Bykovsky Peninsula (6.6% $\pm$ 7.4) have a higher TOC content than samples from Sobo-Sise Island (3.5% $\pm$ 3.8). Differentiating into the various landform types, samples from thermokarst on Sobo-Sise store less TOC (3.6% $\pm$ 3.9) than samples from Yedoma uplands (4.3% $\pm$ 4.2) which is contrasting with the Bykovsky site (7.9% $\pm$ 9.0 for thermokarst and 5.1% $\pm$ 4.3 for Yedoma upland). However, the standard deviations and therefore the variance are higher on the Bykovsky Peninsula thermokarst samples.

The total nitrogen (TN) contents show a similar pattern like the TOC values. There is less TN in the samples from Sobo-Sise. There are, however, only very small differences in TN between the different landform types for both study areas.

Whereas there is slightly more TN in Yedoma upland samples than thermokarst samples on Sobo-Sise, there is less in Yedoma uplands compared to thermokarst on Bykovsky Peninsula. However, when looking at the median, the sample series (Yedoma upland on Sobo, Yedoma upland on Bykovsky and thermokarst on Bykovsky) tend to be similar (see Table S2 with the median values in the supplementary material).

Volumetric ice contents are very similar for all the sampled cores. For both study sites as well as for Yedoma and thermokarst the mean values range between 61% and 67%. The only exceptions are the samples from fluvial sandy deposits on Sobo-Sise Island with a mean value of 45% (see Table 1). Active layer samples were excluded from this analysis.

CN ratios also show a similar pattern across all classes, with mean CN ratios ranging from 10.7 to 13.2 for the different geomorphological units. However when separating CN ratios into active layer and permafrost layer samples, active layer samples show a higher CN ratio than samples in the permafrost layer at both study sites and in all classes. Mean CN ratios for active layer samples (samples from cores of same landform units combined) are: 15.8; 20.1; 12.5; 17.0 for Sobo-Sise Yedoma uplands, Sobo-Sise Thermokarst, Bykovsky Yedoma Upland and Bykovsky Thermokarst, respectively. For permafrost samples mean CN ratios are: 11.1; 10.3; 10.1; 11.7 for Sobo-Sise Yedoma uplands, Sobo-Sise Thermokarst, Bykovsky Yedoma Upland and Bykovsky Thermokarst, respectively. In all cases, active layer samples have higher CN ratios and there is a moderate correlation of decreasing CN ratio with increasing depth for three classes, and one weak correlation (Bykovsky Thermokarst) of decreasing CN ratio with increasing depth (Fig. 2).

**3.2 Sampling site SOC and N stocks**

The TOC and bulk density values were used to estimate the site SOC stocks which were averaged for the different landform types. Mean sampling site SOC stocks (excluding ice wedge volume) were higher for Yedoma upland sites than for thermokarst sites for the reference depths of 0-30 cm and 0-100 cm for both study sites (Fig. 3). SOC storages for 0-100 cm for Sobo-Sise are $25.3 \pm 8.0$ kg C m$^{-2}$ for Yedoma upland and $19.2 \pm 5.9$ kg C m$^{-2}$ for thermokarst sites. For fluvial deposits only one profile down to one meter depth was collected with an SOC stock of 11.2 kg C m$^{-2}$. Also for 0-200 cm more SOC is stored in Yedoma upland soils. The findings for Bykovsky Peninsula are similar with more C stored in Yedoma uplands sites in the first meter of soil than in thermokarst sites, despite the fact that a higher mean TOC content was found in the thermokarst samples. For 0-100 cm Yedoma upland sites store $29.7 \pm 12.9$ kg C m$^{-2}$ and thermokarst sites store $23.9 \pm 9.7$ kg C m$^{-2}$. For 0-200 cm there is more organic C stored in thermokarst than in Yedoma upland soils. However, this estimation is only based on one relatively C rich core (BYK14-T3-3), since this is the only core reaching a depth of two meters for thermokarst on Bykovsky Peninsula. Therefore the carbon estimation for thermokarst on Bykovsky Peninsula for the soil interval 0-200 cm has to be interpreted carefully.

The mean sampling site soil N stock (excluding ice wedge volume) for Yedoma upland sites on Sobo-Sise Island is $2.3 \pm 1.1$ kg N m$^{-2}$ (0-100 cm) and for thermokarst sites $1.4 \pm 0.4$ kg N m$^{-2}$ (Fig. 4). Mean profile N storage for the first meter of soil on Bykovsky Peninsula are 2.6 kg N m$^{-2}$ $\pm$ 0.9 for Yedoma upland sites and 1.9 kg N m$^{-2}$ $\pm$ 0.6 for thermokarst sites.

### 3.3 Upscaling: Landscape SOC and N stocks

The landform classification (Fig 5.) shows that 43% and 51% of the landscape on Sobo-Sise and Bykovsky, respectively, are Yedoma or partly degraded Yedoma. Thermokarst depressions cover approximately 43% on Sobo-Sise and 38% on Bykovsky (excluding lakes and lagoons which cover 14% and 11% of the landscapes, respectively).

Based on the landform classification and the sampling site C contents the total C storage for the two study regions was calculated. In total, 5.81 Tg of organic C are stored in the first meter of soil on Sobo-Sise (288 km$^2$) of which around 57% are stored within the active layer. These calculations include a landscape-wide ice wedge volume of 40% for Yedoma and 9% for thermokarst, always applied for deposits below the active layer. This results in an average SOC storage for non-lake areas on Sobo-Sise of 20.2 ± 2.9 (95% confidence interval) kg C m$^{-2}$ for 0-100 cm. For Bykovsky Peninsula (154 km$^2$) the results are similar. In the first meter of soil, 3.98 Tg of organic C are stored of which 58% are stored in the active layer, including an ice wedge volume of 44% for Yedoma and 9% for thermokarst. This results in a landscape average of 25.9 ± 9.3 kg C m$^{-2}$ (excluding lakes) for 0-100 cm.

Beside the organic C, there is a considerable amount of N stored in the soils of Sobo-Sise Island and Bykovsky Peninsula. About 0.53 Tg of N is stored on Sobo-Sise resulting in a mean N storage of 1.8 ± 0.2 kg N m$^{-2}$ (0-100 cm, excluding lakes). On Bykovsky, a total of 0.34 Tg of N are stored in the first meter of soil. This results in a mean N stock of 2.2 ± 0.5 kg N m$^{-2}$ for 0-100 cm. Mean N and organic C storage for the reference depths and the two study areas are summarized in Table 2 and the total landscape stocks are presented in the supplement Table S3.

### 3.4 Radiocarbon dates

In general, the radiocarbon dates (Table 3) indicate that organic matter in the first two meters (and in one thermokarst site down to three meters) in both study areas predominantly is of Holocene age. Only one Yedoma upland site (BYK14-T3-6B) has clearly late Pleistocene ages around one meter depth. The two other dated Yedoma upland sites BYK14-T2-3 and SOB14-T2-2 indicate the presence of a thick Holocene cover layer exceeding 1.67 m and 2.23 m, respectively. In general, only one age-depth inversion was found (SOB14-T2-5), at all other sites, organic matter age increased with depth.

In addition, cumulative SOC storage and radiocarbon dates were combined to calculate organic C accumulation rates (Table 4). Figure 6 shows the radiocarbon ages plotted against the cumulative SOC for each sampling site, indicating the C accumulation rates. The plots show that the C accumulation rate was fairly linear in all of the cores, especially when removing one age-inversion from a core (SOB14-T2-5 with 5,517 cal yr BP) and one outlier (the exceptional old date from BYK14-T2-3 with 45,203 cal yr BP), however the accumulation rates vary for both the two classes thermokarst and Yedoma upland soils as well as for single sampling sites. The highest mean SOC accumulation rate is found in the thermokarst site SOB14-T2-5 with 49.7 g C m$^{-2}$ yr$^{-1}$ (and 300 cm of sediment accumulation) which is almost 20 times higher than SOB14-T1-5, another thermokarst site from Sobo-Sise which has a mean organic C accumulation rate of 2.7 g C m$^{-2}$ yr$^{-1}$ (and 200 cm of sediment accumulation). This is also reflected in the different sediment accumulation rates (Table 4). This high variability in sediment

and C accumulation rates reveals that even within a small area (the sites are located within 3 km) a high heterogeneity exists in soil forming and C accumulation processes.

## 4 Discussion

### 4.1 Site specific soil organic C and N stock characteristics

We found that particularly DTLBs contain less C than what has been estimated by other studies (Strauss et al., 2013; Walter Anthony et al., 2014). However, Strauss et al. (2013) and Walter Anthony et al. (2014) included also samples from greater soil depths and partially included sites from boreal regions with higher net primary productivity, whereas our study was focusing on the first three meters of the soils in a high latitude tundra region. Strauss et al. (2013) did a C inventory for the entire Yedoma region resulting in a SOC storage of 10 + 17/-6 kg m$^{-3}$ for Yedoma and 31 + 23/-18 m$^{-3}$ for thermokarst deposits.

Especially thermokarst sites on Sobo-Sise Island are more depleted in SOC and store less N than the Yedoma upland sites. In contrast to several previous studies investigating drained thermokarst lake basin peat accumulation in Alaska, (Bockheim et al., 2004 Hinkel et al., 2003; Jones et al., 2012) and the Kolyma region in Siberia (Walter Anthony et al 2014), it is clear that the investigated DTLB soils in Sobo-Sise and Bykovsky do not show signs of increased peat formation and contain only thin organic layers. Organic layer depths of the studied DTLBs on Sobo-Sise and Bykovsky Peninsula are largely

less than 10 cm with only two sites having thicker organic-rich peaty layers in the top. This indicates that the conditions for peat accumulation in these DTLB were not favorable. Also for Yedoma upland soils, organic layers are relatively shallow (< 10 cm). This is especially important when considering that the organic layer which insulates the ice rich Yedoma deposits from warming and thawing is only thin, rendering the Yedoma in this region vulnerable to active layer deepening and permafrost degradation. Due to the lack of thick organic or peaty layers, most of the profiles were classified as the mineral-dominated

Orthels or Turbels. This is consistent with the argumentation in Hugelius et al. (2016) which emphasized that DTLBs do not always contain peaty C rich deposits (Histels).

A key reason for the rather low SOC contents in DTLBs is the low primary productivity of the study sites at ~72° N latitude. In addition, the sampled DTLB represent only a fraction of all the basins in the study areas and may not be representative of the full range of basin ages. Previous studies from Alaska indicated that older basins contained thicker organic

layers than younger basin (Hinkel et al. 2003; Jones et al., 2012). Other reasons may include the topographic gradient that impacts how well the DTLB are drained and whether these basins remain waterlogged peat-forming landscapes or become dry environments not favorable for peat formation. Several of the studied DTLB were eroded by the Lena River (Sobo-Sise) or the sea (Bykovsky Peninsula) and some have deeply incised drainage channels, all of which caused rather strong drainage gradients and enhanced landscape drying.

Our mean landscape SOC stocks for Sobo-Sise Island and Bykovsky Peninsula are, however, in the same range of previous studies in similar settings. E.g. Siewert et al. (2016) found a mean of 19.2 kg C m$^{-2}$ (0-100 cm) in another Yedoma dominated landscape in the central Lena River Delta. Zubrzycki et al. (2013) investigated the SOC characteristics of the

Holocene river terrace and the active floodplain in the Lena River Delta and found mean SOC stocks of 29.5 kg C m$^{-2}$ and 13.6 kg C m$^{-2}$, respectively. The C storage of the active floodplain is therefore very similar to the 11.2 kg C m$^{-2}$ for fluvial deposits in our study area. Ping et al. (2011) also investigated C storage along the Alaska Beaufort Sea coastline and found a normalized mean landscape storage of 38 kg C m$^{-2}$. However, this number is based on profiles only and not on a landscape

based upscaling. Shmelev et al. (2017) investigated sites in the Kolyma Yedoma region and found a C storage of 17.0 ± 51.1 kg C m$^{-3}$ (note that it is kg C m$^{-3}$) for the Holocene cover layer, 16.2 ± 31.3 kg C m$^{-3}$ for the Alas (thermokarst) deposits and 14.0 ± 23.5 kg C m$^{-3}$ for Yedoma deposits. These values however refer not only to the first two meters of soil covering Pleistocene Ice Complex deposits but to the upper 25 m of Yedoma Ice Complex. Also, Webb et al. (2017) investigated deep (15 m) C stocks in a larch dominated Yedoma area in the Kolyma river basin and found more organic C in the Alas site than

in the Yedoma site. Based on a landscape upscaling, Siewert et al. (2015) calculated the SOC storage for the Kytalik region, a Yedoma and thermokarst dominated tundra in the Yana-Indigirka Lowland, to 25.8 ± 9.9 kg C m$^{-2}$ for the first meter of soils which falls in the range spanned by the Sobo-Sise Island and Bykovsky Peninsula calculations.

Aside from organic C, a significant amount of N is stored in the soils of Bykovsky Peninsula and Sobo-Sise Island, almost twice as much as what has been found by the study from Zubrzycki et al. (2013) for the Holocene river terrace (1.2 kg

m$^{-2}$) and the active floodplain (0.9 kg m$^{-2}$) of the Lena Delta. Obu et al. (2015) reported higher N storages from the western Canadian Arctic, where 3.4 kg N m$^{-2}$ are stored in cryoturbated or recent disturbed type soils or sediments and Michaelson et al. (2013) calculated a mean N storage of 2.7 kg m$^{-2}$ for Arctic Alaska pedons. The normalized average N storage from the Alaskan Beaufort Sea coast is in the same range with 1.9 kg N m$^{-2}$ (Ping et al., 2011) and also N storages from two study sites on Taymir Peninsula are in the same range, with 1.0 and 1.3 kg N m$^{-2}$ (Palmtag et al., 2016).

Even though the N storages are an order of magnitude lower than the organic C storages, a large amount of N is present in these soils. Since N is the limiting factor for plant growth in Arctic environments (Shaver et al., 1986; Chapin et al., 1995; Mack et al., 2004; Beermann et al., 2015), permafrost thawing will affect the N stocks in the soils. The N could partially become available to plants upon permafrost thawing. However, the role of N and whether it can offset an increased organic carbon release through increased plant growth needs further exploration. In a recent study, Keuper et al. (2017) found that

plant available N from thawing permafrost is an additional source for deep-rooting subarctic plants and can increase their biomass production. Also Salmon et al. (2016) reported that increased N from thawed permafrost enhances plant growth and biomass, however that it might not offset C from deep deposits. An increase in N has, however, the potential to change the plant productivity and the species composition (Keuper et al., 2012). Keuper et al. (2012) shows that not only dissolved plant available N becomes available with permafrost thawing but also organically bound N can be mineralized at faster rates in

thawed near permafrost soils.

However, an increased N availability might stimulate both, vegetation growth but also increase microbial activity (Nowinski et al. 2008). Also, Koven et al. (2015) reports that near surface N released from thawing permafrost might reduce nutrient limitations, even though the same study shows limited importance of the deep soil N to offset deep C release. In an expert assessment Abbott et al. (2016) stated that even an increase in Arctic and boreal biomass might not offset permafrost

carbon release. Since we only focused on N stocks, we cannot derive conclusions on potential plant available N in the soils of Sobo-Sise Island and Bykovsky Peninsula and its effect on the primary production or C release. Nevertheless, this first estimation of total N in the soils of Sobo-Sise Island and Bykovsky Peninsula will be relevant for future climate models.

## 4.2 Upscaling of C and N pools

The total C stocks of our study sites are in the range of other permafrost C studies and confirm previous high C stock estimates from northern permafrost regions (e.g. Hugelius et al., 2014). The landform classification proved to be adequate for the upscaling. The overall accuracy for the classification is 71.5% and 71.1% for Sobo-Sise and Bykovsky, respectively. Additional field data in more different sub-classes and a more diversified classification would further increase the precision of the upscaling. Nevertheless, by including a high resolution DEM, classifying remotely sensed images into geomorphological
landform types results in an accurate map for a first estimation of SOC and N stocks on the landscape level. In this context, of particular note is the areal fraction of Yedoma uplands we found on Sobo-Sise Island and the Bykovsky Peninsula of 43% and 51%, respectively. While this is within the range Morgenstern et al. (2011) described for the 3rd terrace of the Lena Delta, on a larger regional level Yedoma coverage is generally lower. For example Veremeeva and Glushkova (2016) calculated 16% of Yedoma coverage for the entire Kolyma Lowland. However, our, higher values are a combination of Yedoma uplands and
partly degraded Yedoma slopes. Excluding areas with slopes, 19% of Sobo-Sise and 22% of Bykovsky Peninsula are covered by intact Yedoma uplands not yet significantly affected by thermokarst or erosion.

For upscaling deep C and N pools, the determination of ice wedge volumes is important. In our study we assessed ice wedge volume by a combination of literature values and own values derived form a GIS based analysis of high resolution satellite data. This analysis was based on more than 1500 mapped baydzheraks and resulted in a mean ice wedge volume of
20 40% for Sobo-Sise Island with a standard deviation of ± 8%. Conducting additional calculations with an ± 8% ice wedge volume for Yedoma uplands and ± 3% for thermokarst areas would lead to a ± 4% higher respective lower landscape SOC stock (± 5% for N) for 0-100 cm and ± 7% of SOC and total N for 0-200 cm for Sobo-Sise Island. While these results are still in the same range, this analysis reveals the importance to assess ice wedge volumes correctly within a landscape for SOC and N upscaling, especially for deep deposit calculations. For a more detailed upscaling and to capture the entire variability of
25 these heterogeneous environments, additional and deeper soil cores are needed as well as a more detailed estimation of the landscape ice wedge contents will further improve SOC and soil N pool estimations.

## 4.3 Sediment and organic C accumulation rates

Most of the analyzed soil C was of Holocene age. Even for Yedoma upland soils radiocarbon dates indicated a large number of Holocene ages. Hence, the Yedoma uplands appear to be blanketed by Holocene cover material sometimes
exceeding a thickness of two meters, which needs to be accounted for in carbon pool inventories, because these two meter cover deposits cannot be considered as Yedoma. In DTLBs, the Holocene age of soil C fits well with the findings from previous studies suggesting that the accumulation of lacustrine sediments, drainage of thermokarst lakes, and accumulation of soils and

organic layers in the basins has been occurring mostly during the Holocene (Kaplina 2009; Grosse et al., 2013; Walter Anthony et al., 2014). Based on the radiocarbon dates and the cumulative SOC storages, the accumulation rates for the soil cores were calculated. The mean (linear) C accumulation rate of SOB14-T2-5 with 49.7 g C m$^{-2}$ yr$^{-1}$ is very high, even higher than what has been found by Jones et al. (2012) for paleo peat accumulation rates (9 – 35.2 g C m$^{-2}$ yr$^{-1}$) in thermokarst basins on the

Seward Peninsula. Nevertheless, it has to be considered that this site location is close to the Lena River with only approximately 5 m above river water level. Likely, this location is affected by spring flood events which can deposit large amounts of sediments. On the other side, the site SOB14-T1-5 with 2.7 g C m$^{-2}$ yr$^{-1}$ has very low accumulation rates and is most certainly not affected by the Lena River flood. For comparison, Kurganova et al. (2014) find that modern C accumulation on arable land in Russia was on average 9.6 g C m$^{-2}$ yr$^{-1}$ over a 20 year period after abandonment, Hicks Pries et al. (2012) found a mean

Holocene C accumulation rate of 25.8 g C m$^{-2}$ yr$^{-1}$ for surface soils and 2.3 g C m$^{-2}$ yr$^{-1}$ for deep soils in subarctic tundra in central Alaska, and Bockheim et al. (2004) found a mean long-term accumulation rate of 13 g C m$^{-2}$ yr$^{-1}$ for the last 5,500 years in DTLBs in Arctic Alaska, however referring to the organic layer only.

Sediment accumulation rates are all in the same order of magnitude, excluding SOB14-T2-5, which might be affected by the Lena River, with mean accumulation rates for the different thermokarst cores ranging between 0.13 mm yr$^{-1}$ and 0.26

15  mm yr$^{-1}$ and mean accumulation rates for Yedoma upland cores between 0.10 mm yr$^{-1}$ and 0.57 mm yr$^{-1}$ (Table 4). For comparison, Murton et al. (2015) found sediment accumulation rates between 0.75 and 2.00 mm yr$^{-1}$ for Yedoma silt at Duvanny Yar in the Kolyma Lowland. However, these rates refer to a time period between 38,700 and 23,600 years BP. Also Schirrmeister et al. (2002a, 2002b) found a similar accumulation rate for a Yedoma deposit (Mamontovy Khayata) on Bykovsky Peninsula of about 0.75 mm yr$^{-1}$ for the time period between 60,000 to 6,000 years BP. These rates are slightly

higher than the rates calculated in our study. In addition, all presented sediment accumulation rates (Table 4) will be lower when taking into account the ice content within the deposits. Sediment accumulation rates corrected for soil core ice content are presented in the Supplement Table S6.

## 4.4 Characterizing soil organic carbon

The rather low CN ratio in our study is common to all sampled soils. Only individual samples showed higher CN ratios and in

general there is a trend of decreasing CN ratios with increasing soil depth. Even though the permafrost organic matter is already partly degraded, these finding suggests that organic matter in the top permafrost layer may be remobilized and decomposed when thawed out as the result of fluctuation of the permafrost table due to climate change. Also Strauss et al. (2015) found relatively small CN ratios for Yedoma and thermokarst samples (median values below 8 and 10), although they looked at samples from deeper deposits. The mean values from our study might be lower too when incorporating samples from greater

depths, this is indicated by the decreasing CN ratio with increasing depth. Higher CN values in the upper meter of soil were found by Zubrzycki et al. (2013) with mean values between 20 and 42 from Holocene river terrace and mean values between 13 and 21 for the active floodplain level of the Lena Delta. This indicates fresher material in these deposits compared to Yedoma uplands and DTLB deposits.

**4.5 The fate of organic carbon in thermokarst-affected Yedoma in Siberia**

Permafrost soil layers beneath the active layer and below one meter depth are important for future C remobilization, because models suggest permafrost degradation and thaw well beyond 1 m depth by end of the 21[st] century (Lawrence et al., 2012; Koven et al., 2013). The cores and the high sample resolution in this study provide detailed information on the C stored in the soil beneath the active layer in the study areas which will be thawed first by future warming. This study provides additional soil C and N data for multiple cores deeper than 100 cm for thermokarst-affected Yedoma landscapes. Studies with such deeper cores are rare and even the NCSCD contains three times more profiles for the 0-100 cm (1778 profiles) interval than for estimations exceeding 100 cm depth (Hugelius et al., 2014).

Our upscaling suggests that the study sites contain significantly more C than soils in temperate climate zones (e.g. Wiesmeier et al., 2012). Both study areas could become sources of organic C and N if permafrost thaw continues in a warming Arctic. An estimation based on the sampled cores and the landform classification shows that with an overall deepening of the active layer of 10 cm 700,000 t C (1.6 kg C m$^{-2}$) will thaw out in both study areas combined (Table 5). A regional study of Siberian permafrost dynamics (Sazonova et al., 2004) includes scenarios where the active layer deepens by more than 100 cm in Northeastern Siberia at the end of the 21[st] century. This would result in an additional pool of available SOC of 5,830,000 t (13.2 kg C m$^{-2}$) in the two study areas combined. In addition, other forms of permafrost thaw than active layer deepening would further increase the amounts of SOC thawed and mobilized. For example, lake shore erosion rapidly degrades permafrost around lakes and releases organic C to the aquatic environment. Shore erosion not only affects the active layer soils from the top but does affect deeper permafrost soil layers (Walter Anthony et al., 2016). This vulnerability of large currently frozen C pools to thaw highlights the importance of deep permafrost organic C to be considered in future C cycle models.

**5 Conclusions**

This study presents the first SOC and N inventories for Sobo-Sise Island and Bykovsky Peninsula, two Yedoma-dominated and thermokarst-affected landscapes in the North of East Siberia for the first two meters of soil. These ice rich permafrost landscapes are vulnerable to climate warming and have the potential to release large amounts of SOC and total N through active layer deepening and permafrost thaw.

Sampling sites in DTLBs were found to contain less organic C than soils on the Yedoma upland. Permafrost soils in DTLBs were all of Holocene age and soils of the upper two meters on the Yedoma uplands largely were all part of a Holocene cover layer above late Pleistocene Yedoma deposits. The mean upscaled landscape SOC storage for 0-100 cm is 20.2 ± 2.9 kg C m$^{-2}$ for Sobo-Sise and 25.9 ± 9.3 kg C m$^{-2}$ for Bykovsky Peninsula which results in a total storage of 9.8 Tg C across both study areas for the first meter of soil. Based on our high sample density, detailed C estimations for active layer deepening were derived, suggesting that 5.8 Tg (13.2 kg C m$^{-2}$) of SOC may become available for microbial degradation due to thaw if the

active layer deepens by 100 cm in the two study areas. The N stocks are one order of magnitude lower, nevertheless a mean of $1.8 \pm 0.2$ kg N m$^{-2}$ are stored on Sobo-Sise and $2.2 \pm 0.5$ kg N m$^{-2}$ on Bykovsky Peninsula for the 0-100 cm depth interval. Therefore, as a consequence of permafrost thaw not only SOC but also additional N may become available for plants and microorganisms.

5         This study confirms the importance of Yedoma and thermokarst landscapes for the permafrost C pool and adds important shallow and deep C and N data to the permafrost region soil databases. It also shows the high variability of SOC and N distribution in thermokarst-affected Yedoma environments. Our study particularly underlines the benefits of soil cores beyond 1 m depth to capture the high variability both within the soil and the entire landscape. Our result of C and N storage and availability to permafrost thaw in the upper subsurface points towards the fact that not only the amount but also the potential for remobilization and the fate of freshly thawed matter requires more attention in the Yedoma region. Even though Sobo-Sise Island and the Bykovsky Peninsula do not contain extraordinary high SOC stocks in the near-surface permafrost at profile level, they nevertheless have a large potential for rapid mobilization of significant amounts of C due to their large proportion of thaw-vulnerable juvenile Yedoma and thermokarst-affected landscape units.

Supplementary data are available at https://doi.pangaea.de/10.1594/PANGAEA.883582.

15 **Author contributions**

M.F designed the study; M.F., G.G., J.S., F.G., M.G., G.M.M conducted the field work during the expedition Lena 2014; M.F. carried out the laboratory analysis; F.G. generated the high resolution DEMs; M.F wrote the publication with input from all co-authors.

**Acknowledgements**

20 This study was supported by ERC Starting Grant #338335 and the Initiative and Networking Fund of the Helmholtz Association (#ERC-0013). We thank T. Opel and I. Nitze for the help with coring on Sobo-Sise Island and Bykovsky Peninsula during the expedition Lena Delta 2014, and S. Jock, T. Henning, and D. Scheidemann for help with the laboratory work. RapidEye imagery was kindly provided by the German Aerospace Center (DLR) and BlackBridge AG through the RapidEye Science Archive. The authors thank the editor and three anonymous reviewers for their constructive comments.

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

**Tables**

**Table 1**: Laboratory results of soil C, N, and ground ice characteristics for the different geomorphological landform units on Sobo-Sise Island and Bykovsky Peninsula. All values are mean values ± standard deviation. Median values for all parameters are provided in the supplement material in Table S2.

| | TOC [%] | TIC [%] | TN [%] | C/N [-] | Vol. ice content[a] [%] | SOC [kg/m$^3$] | N [kg/m$^3$] | n |
|---|---|---|---|---|---|---|---|---|
| **Sobo-Sise** | 3.5 ± 3.8 | 0.2 ± 0.4 | 0.3 ± 0.1 | 10.9 ± 6.3 | 61.8 ± 14.8 | 19.5 ± 12.6 | 1.7 ± 0.9 | 279 |
| **Sobo-Sise Yedoma upland** | 4.3 ± 4.2 | 0.3 ± 0.4 | 0.3 ± 0.1 | 12.2 ± 6.8 | 61.5 ± 15.0 | 25.7 ± 16.1 | 2.2 ± 1.1 | 85 |
| **Sobo-Sise Yedoma slope** | 3.1 ± 1.8 | 0.1 ± 0.1 | 0.3 ± 0.1 | 10.7 ± 2.9 | 61.6 ± 16.8 | 19.6 ± 10.0 | 1.8 ± 0.7 | 33 |
| **Sobo-Sise Thermokarst** | 3.6 ± 3.9 | 0.1 ± 0.4 | 0.3 ± 0.1 | 11.7 ± 5.7 | 63.1 ± 13.7 | 17.6 ± 9.4 | 1.6 ± 0.6 | 135 |
| **Sobo-Sise Fluvial Deposits** | 0.8 ± 0.7 | 0.0 ± 0.1 | 0.1 ± 0.0 | 10.8 ± 6.0 | 45.0 ± 9.4 | 10.0 ± 7.2 | 0.9 ± 0.5 | 26 |
| **Bykovsky** | 6.6 ± 7.4 | 0.7 ± 0.4 | 0.5 ± 0.3 | 11.9 ± 6.5 | 62.8 ± 16.4 | 28.9 ± 21.2 | 2.4 ± 1.2 | 176 |
| **Bykovsky Yedoma upland** | 5.1 ± 4.3 | 0.6 ± 0.4 | 0.4 ± 0.2 | 10.8 ± 3.5 | 62.1 ± 13.6 | 28.1 ± 18.4 | 2.5 ± 1.2 | 80 |
| **Bykovsky Thermokarst** | 7.9 ± 9.0 | 0.8 ± 1.0 | 0.5 ± 0.4 | 13.2 ± 7.8 | 66.4 ± 14.3 | 29.5 ± 23.3 | 2.2 ± 1.3 | 96 |

5  [a] Intrasedimentary ice (excluding wedge ice volume)

**Table 2:** Mean landscape organic C and N storages in kg C m$^{-2}$ ± 95% confidence interval

| Study site | SOC active layer | SOC 0-30 cm | SOC 0-100 cm | SOC 0-200 cm | N active layer | N 0-30 cm | N 0-100 cm | N 0-200 cm |
|---|---|---|---|---|---|---|---|---|
| **Sobo-Sise Island** | 11.6 ± 1.94 | 9.0 ± 1.23 | 20.2 ± 2.91 | 31.3 ± 3.56 | 1.0 ± 0.23 | 0.7 ± 0.09 | 1.8 ± 0.16 | 3.0 ± 0.23 |
| **Bykovsky Peninsula** | 15.0 ± 3.10 | 10.8 ± 2.25 | 25.9 ± 9.33 | 48.4 ± 9.97 | 1.3 ± 0.21 | 0.9 ± 0.13 | 2.2 ± 0.45 | 4.0 ± 0.37 |

**Table 3**: Radiocarbon dates for selected soil cores

| Sample ID | Depth [cm] | Lab. no. | AMS [14]C age [yrs BP] | Calib age [yrs BP][a] | Dated material | Weight [mg] | Latitude [°] | Longitude [°] |
|---|---|---|---|---|---|---|---|---|
| **Yedoma upland** | | | | | | | | |
| SOB14-T1-1-3 | 20-24 | Poz-74518 | 970 ± 30 | 842 ± 46 | Sedge stems | 13 | 72.50442 | 128.03915 |
| **DTLB** | | | | | | | | |
| SOB14-T1-5-2 | 10-11 | Poz-74451 | 112.88 ± 0.32 pMC | modern | Moss leaves/stems | 12 | 72.50964 | 128.03435 |
| SOB14-T1-5-15 | 148-150 | Poz-74452 | 4460 ± 35 | 5058 ± 91 | Bulk organic | 12 | 72.50964 | 128.03435 |
| SOB14-T1-5-19 | 187-200 | Poz-74454 | 6605 ± 30 | 7481 ± 44 | Moss leaves/stems | 10 | 72.50964 | 128.03435 |
| **Yedoma upland** | | | | | | | | |
| SOB14-T2-2-7 | 55-65 | Poz-74455 | 1420 ± 30 | 1329 ± 39 | Bark of a twig | 12 | 72.52853 | 127.97281 |
| SOB14-T2-2-16 | 119-122 | Poz-74519 | 3065 ± 35 | 3272 ± 92 | Sedge stems | 17 | 72.52853 | 127.97281 |
| SOB14-T2-2-24 | 173-179 | Poz-74538 | 6200 ± 50 | 7114 ± 137 | Wood with bark | 22 | 72.52853 | 127.97281 |
| SOB14-T2-2-30 | 218-223 | Poz-74522 | 8800 ± 50 | 9807 ± 154 | Bulk organic | 20 | 72.52853 | 127.97281 |
| **DTLB** | | | | | | | | |
| SOB14-T2-5-2 | 5-6 | Poz-74523 | 111.54 ± 0.33 pMC | modern | Moss leaves/stems | 28 | 72.52852 | 127.98176 |
| SOB14-T2-5-10 | 67-74 | Poz-74524 | 135 ± 30 | 168 ± 110 | Bulk organic | 25 | 72.52852 | 127.98176 |
| SOB14-T2-5-19 | 145-156 | Poz-74525 | 350 ± 30 | 364 ± 49 | Sedge stems | 13 | 72.52852 | 127.98176 |
| SOB14-T2-5-31 | 273-278 | Poz-74526 | 4735 ± 40 | 5517 ± 70 | Bulk organic | 20 | 72.52852 | 127.98176 |
| SOB14-T2-5-34 | 299-303 | Poz-74857 | 970 ± 30 | 842 ± 46 | Deciduous leaves | 13 | 72.52852 | 127.98176 |
| **Baydzerakh** | | | | | | | | |
| BYK14-T2-3-2b | 14-16 | Poz-74732 | 595 ± 30 | 615 ± 37 | Sedge stems/leaves | 15 | 71.86050 | 129.29276 |
| BYK14-T2-3-4 | 40-45 | Poz-74733 | 1155 ± 30 | 1109 ± 67 | Sedge stems | 21 | 71.86050 | 129.29276 |
| BYK14-T2-3-8 | 68-75 | Poz-74734 | 1670 ± 30 | 1576 ± 53 | Sedge stems | 40 | 71.86050 | 129.29276 |
| BYK14-T2-3-19 | 159-167 | Poz-74735 | 2715 ± 30 | 2812 ± 52 | Sedge stems | 31 | 71.86050 | 129.29276 |
| BYK14-T2-3-20 | 178-179 | Poz-74737 | 41600 ± 1400 | 45203 ± 2512 | Large wood piece | 118 | 71.86050 | 129.29276 |
| **DTLB** | | | | | | | | |
| BYK14-T2-4-4 | 23-24 | Poz-74738 | 600 ± 30 | 615 ± 38 | Sedge stems | 15 | 71.86143 | 129.29530 |
| BYK14-T2-4-10 | 45-48 | Poz-74739 | 1250 ± 30 | 1222 ± 51 | Sedge stems | 16 | 71.86143 | 129.29530 |
| BYK14-T2-4-16 | 76-79 | Poz-74740 | 1545 ± 30 | 1449 ± 78 | Sedge stems | 22 | 71.86143 | 129.29530 |
| BYK14-T2-4-22 | 117-125 | Poz-74741 | 8350 ± 50 | 9368 ± 119 | Plant remains | 60 | 71.86143 | 129.29530 |
| **Yedoma upland** | | | | | | | | |
| BYK14-T3-6B-14 | 110-116 | Poz-89712 | 12990 ± 70 | 15533 ± 256 | Sedge stems | 20 | 71.82236 | 129.31537 |
| BYK14-T3-6B-18 | 142-148 | Poz-89713 | 13350 ± 70 | 16048 ± 225 | Bulk organic | 16 | 71.82236 | 129.31537 |
| BYK14-T3-6B-23 | 185-191 | Poz-89714 | 14770 ± 70 | 17970 ± 215 | Bulk organic | 20 | 71.82236 | 129.31537 |

[a]calibrated with Calib 7.1 software (Stuiver et al., 2017)

**Table 4:** Sediment (sedim. rate) and organic carbon accumulation rates (OC acc. rate). Sediment accumulation rates are based on the depth of the sample and the calibrated radiocarbon date. Organic carbon accumulation rates are based on cumulative soil organic carbon (Cum SOC) storage at a specific depth and the calibrated radiocarbon date at the corresponding depth. Mean sediment and organic carbon accumulation rates are calculated always referring to the soil surface (depth = 0 cm and Cum SOC = 0 kg C m$^{-2}$). Relative sediment and organic carbon accumulation rates are calculated always referring to the sample above a particular sample.

| Sample | Age[a] [cal yr BP] | Cum SOC [kg C m$^{-2}$] | Depth [cm] | Mean sedim. rate [mm yr$^{-1}$] | Relative sedim. rate [mm yr$^{-1}$] | Mean OC acc. rate [g C m$^{-2}$ yr$^{-1}$] | Relative OC acc. rate [g C m$^{-2}$ yr$^{-1}$] |
|---|---|---|---|---|---|---|---|
| SOB14-T1-1-3 | 842 | 10.29 | 19.5-23.5 | 0.26 | 0.26 | 12.2 | 12.2 |
| SOB14-T1-5-2 | modern | 1.21 | 10-11 | na | na | na | na |
| SOB14-T1-5-15 | 5058 | 13.13 | 148-150 | 0.29 | 0.29 | 2.6 | 2.6 |
| SOB14-T1-5-19 | 7481 | 20.34 | 187-200 | 0.26 | 0.18 | 2.7 | 3.0 |
| SOB14-T2-2-7 | 1329 | 18.30 | 55-65 | 0.45 | 0.45 | 13.8 | 13.8 |
| SOB14-T2-2-16 | 3272 | 32.47 | 119-122 | 0.37 | 0.31 | 9.9 | 7.3 |
| SOB14-T2-2-24 | 7113.5 | 43.13 | 173-179 | 0.25 | 0.14 | 6.1 | 2.8 |
| SOB14-T2-2-30 | 9807 | 50.31 | 218-223 | 0.22 | 0.17 | 5.1 | 2.7 |
| SOB14-T2-5-2 | modern | 0.54 | 5-6 | na | na | na | na |
| SOB14-T2-5-10 | 168 | 12.43 | 67-74 | 4.20 | 4.20 | 74.0 | 74.0 |
| SOB14-T2-5-19 | 364 | 23.97 | 145-156 | 4.13 | 4.08 | 65.9 | 58.9 |
| SOB14-T2-5-31 | 5517 | 38.78 | 273-278 | 0.50 | 0.24 | 7.0 | 2.9 |
| SOB14-T2-5-34 | 842 | 41.83 | 298.5-302.5 | 3.57 | 3.14 | 49.7 | 37.4 |
| BYK14-T2-3-2b | 615 | 10.76 | 14-16 | 0.24 | 0.24 | 17.5 | 17.5 |
| BYK14-T2-3-4 | 1109 | 26.07 | 40-45 | 0.38 | 0.56 | 23.5 | 31.0 |
| BYK14-T2-3-8 | 1576 | 43.14 | 68-75 | 0.45 | 0.62 | 27.4 | 36.6 |
| BYK14-T2-3-19 | 2812 | 66.24 | 159-167 | 0.58 | 0.74 | 23.6 | 18.7 |
| BYK14-T2-3-20 | 45203 | 71.34 | 178-179 | 0.04 | 0.004 | 1.6 | 0.1 |
| BYK14-T2-4-4 | 615 | 10.87 | 23-24 | 0.38 | 0.38 | 17.7 | 17.7 |
| BYK14-T2-4-10 | 1222 | 17.63 | 45-48 | 0.38 | 0.38 | 14.4 | 11.1 |
| BYK14-T2-4-16 | 1449 | 29.10 | 76-79 | 0.53 | 1.37 | 20.1 | 50.5 |
| BYK14-T2-4-22 | 9368 | 34.23 | 117-125 | 0.13 | 0.05 | 3.7 | 0.6 |
| BYK14-T3-6B-14 | 15533 | 34.69 | 110-116 | 0.07 | 0.07 | 2.2 | 2.2 |
| BYK14-T3-6B-18 | 16048 | 38.96 | 142-148 | 0.09 | 0.62 | 2.4 | 8.3 |
| BYK14-T3-6B-23 | 17970 | 42.89 | 185-191 | 0.10 | 0.22 | 2.4 | 2.0 |

[a]radiocarbon dates were calibrated with the Calib 7.1 software (Stuiver et al., 2017)

**Table 5:** Potential C thaw out for different active layer deepening scenarios

|  | Active layer + 10 cm | Active layer + 20 cm | Active layer + 50 cm | Active layer + 100 cm | Area |
|---|---|---|---|---|---|
| Sobo-Sise Island | 0.45 Tg | 0.80 Tg | 1.62 Tg | 3.40 Tg | 287.7 km$^2$ |
| Bykovsky Peninsula | 0.25 Tg | 0.48 Tg | 1.32 Tg | 2.44 Tg | 154.0 km$^2$ |
| Total | 0.70 Tg | 1.28 Tg | 2.94 Tg | 5.83 Tg | 441.7 km$^2$ |

**Figures**

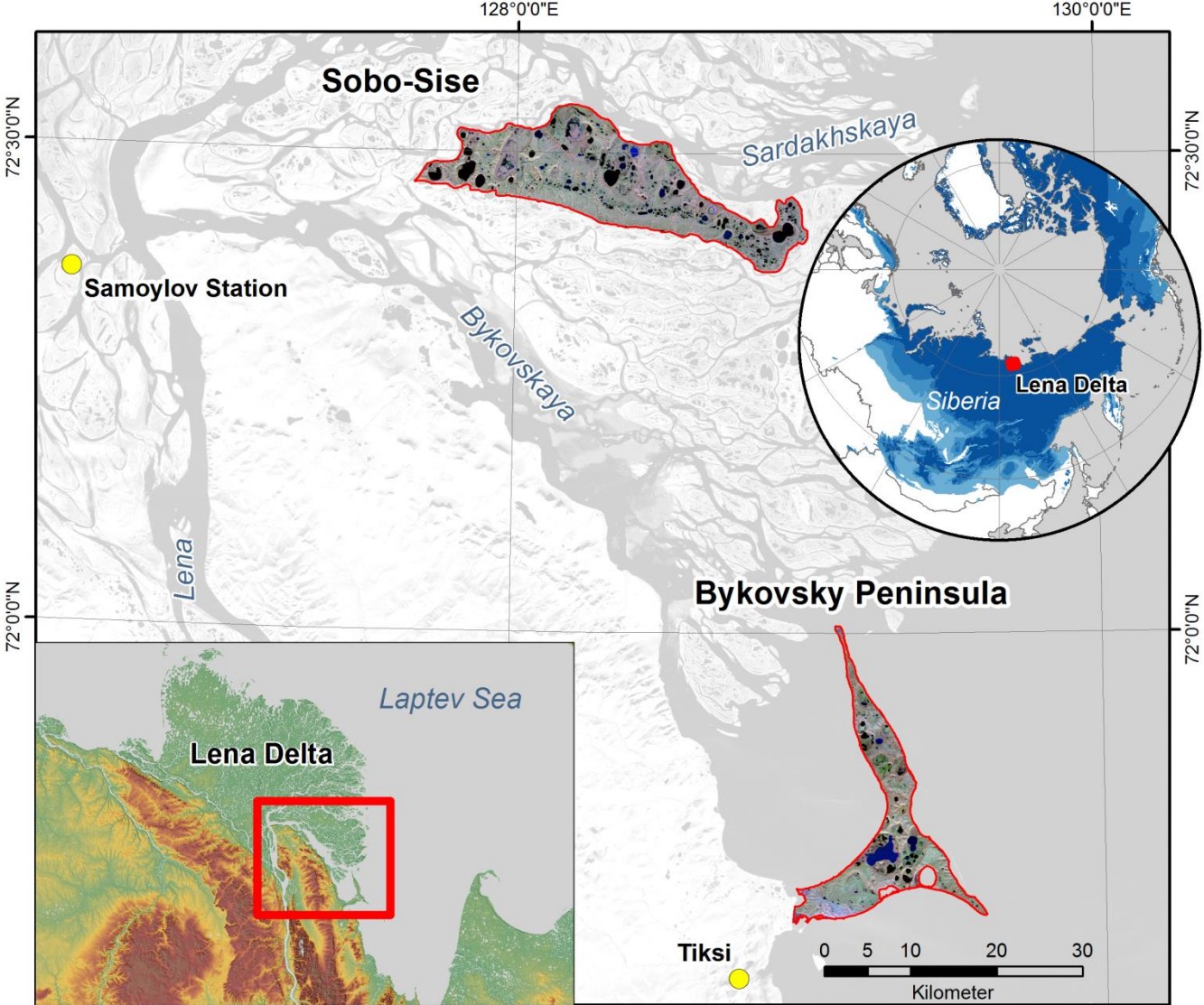

**Figure 1:** Location of the two study areas in the Lena River Delta region, Sobo-Sise Island and Bykovsky Peninsula (Landsat 5 satellite image, acquisition date: 19th Sept. 2009). Top right inset: overview map including the permafrost zonation in Siberia (after Brown et al., 1997), bottom left inset: the Lena Delta region with a digital elevation model (ESA DUE-Permafrost DEM, Santoro and Strozzi, 2012).

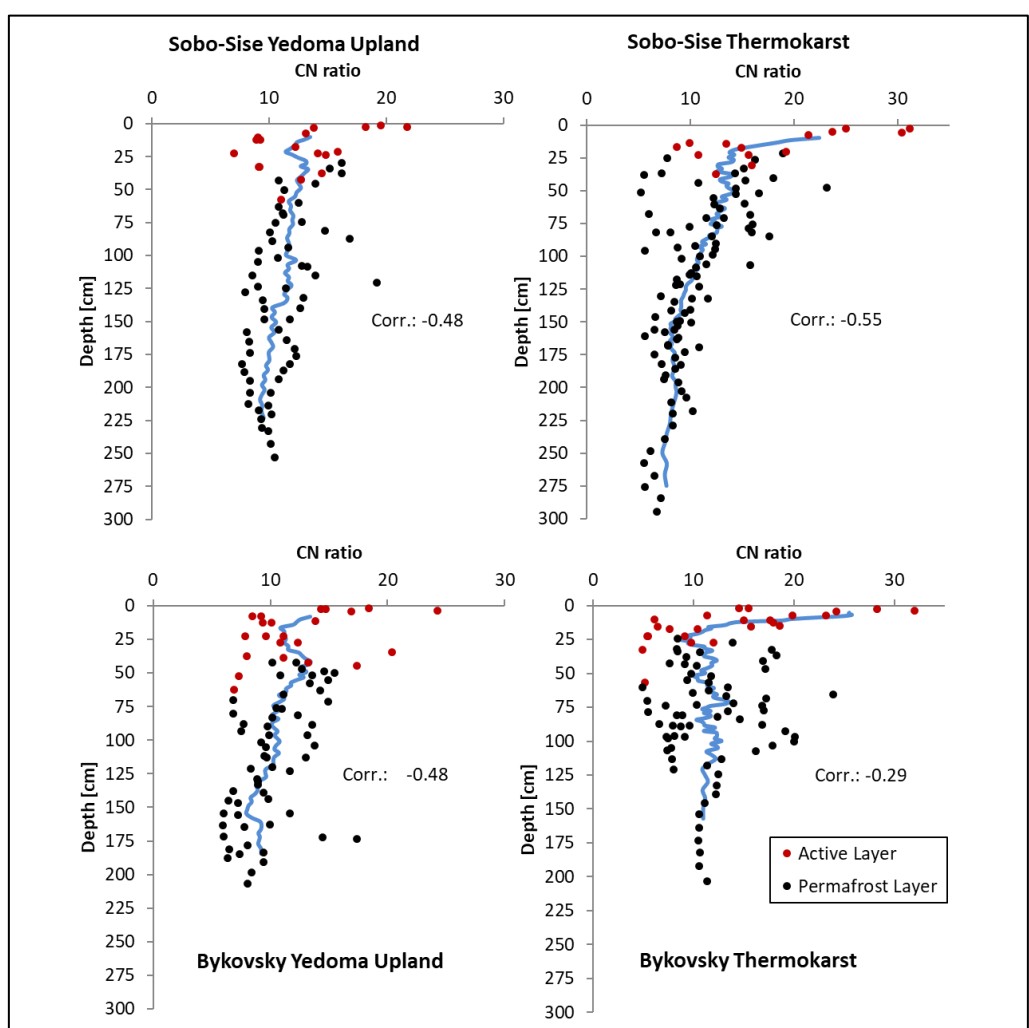

**Figure 2:** CN ratio for Yedoma upland and thermokarst samples on Sobo-Sise and Bykovsky Peninsula. Blue lines indicate the running mean for the entire sample set (including active layer and permafrost layer samples). The correlation (Corr.) (Pearson correlation) between CN ratio and depth indicates a decreasing CN ratio with increasing depth ($p < 0.01$).

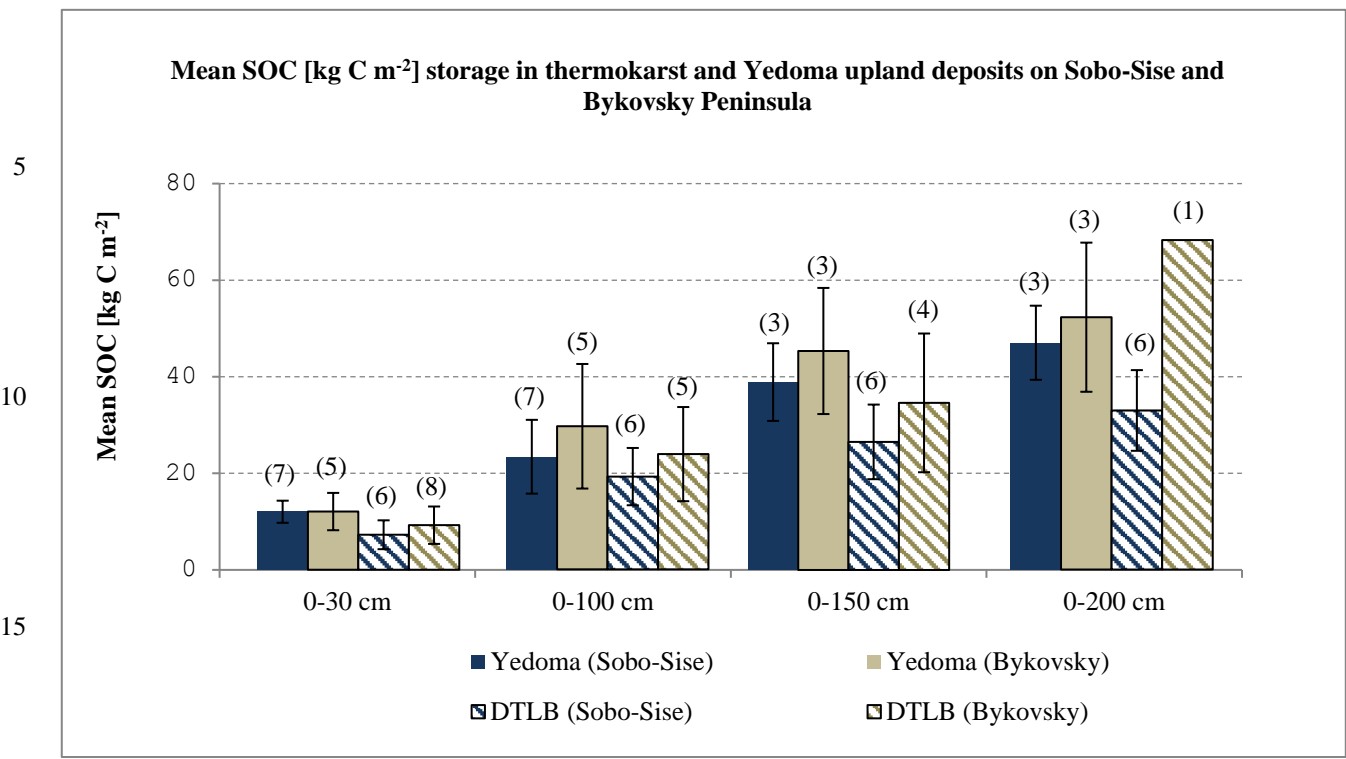

**Figure 3:** Comparison of mean sampling site SOC storage on Sobo-Sise Island and on Bykovsky Peninsula. Solid bars: Yedoma sites, striped bars: Thermokarst sites. Black T-lines show the standard deviation and number in brackets indicate the number of sampled sites. Profiles shorter than 200 cm were extrapolated to the next reference depth. When reaching an ice wedge in a collected site, this was included in the extrapolation as well, assuming no carbon for ice wedge layers. SOC data for this graph are presented in the supplementary material, Table S1 and S4.

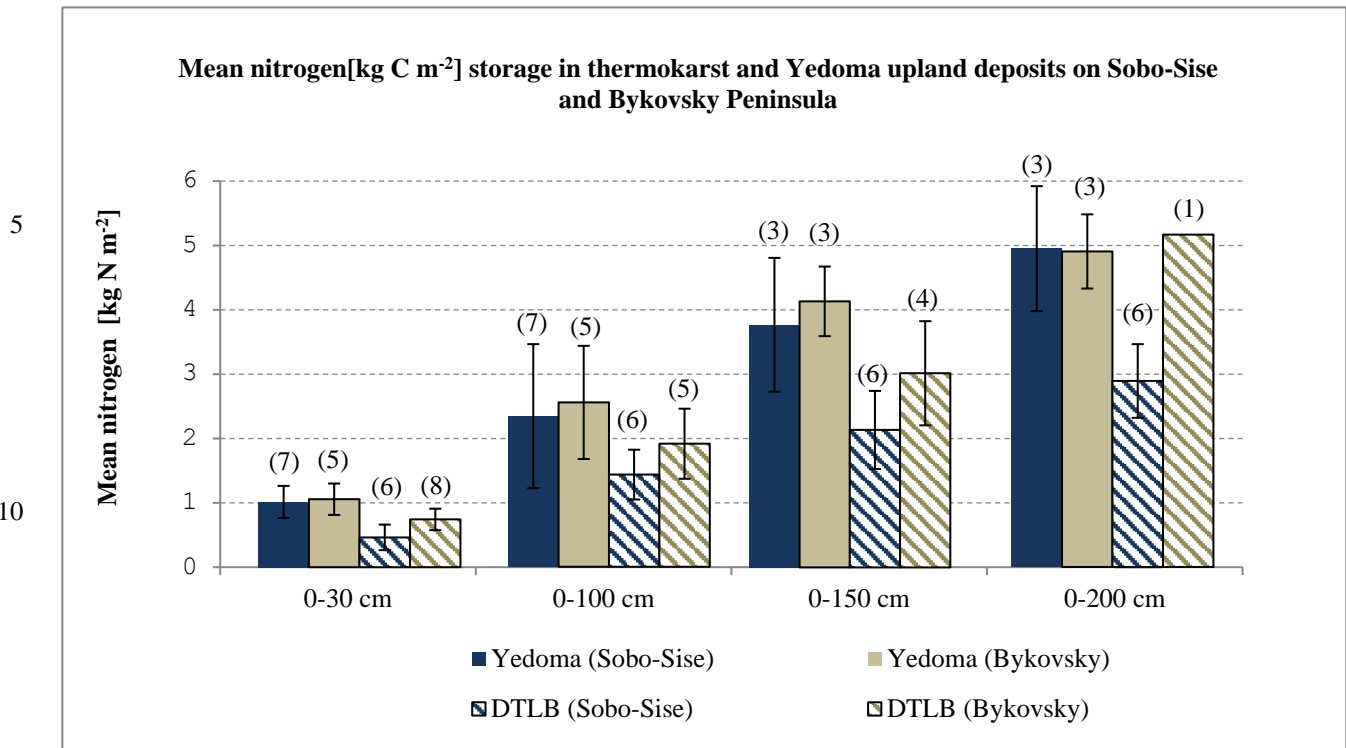

**Figure 4:** Mean sampling site N storage for the different geomorphic units in both study areas. Solid bars: Yedoma sites, striped bars: Thermokarst sites. Black T-lines show the standard deviation and number in brackets indicate the number of sampled sites. Profiles shorter than 200 cm were extrapolated to the next reference depth. When reaching an ice wedge in a collected site, this was included in the extrapolation as well, assuming no N for ice wedge layers. Soil N data for this graph are presented in the supplementary material, Table S1 and S5.

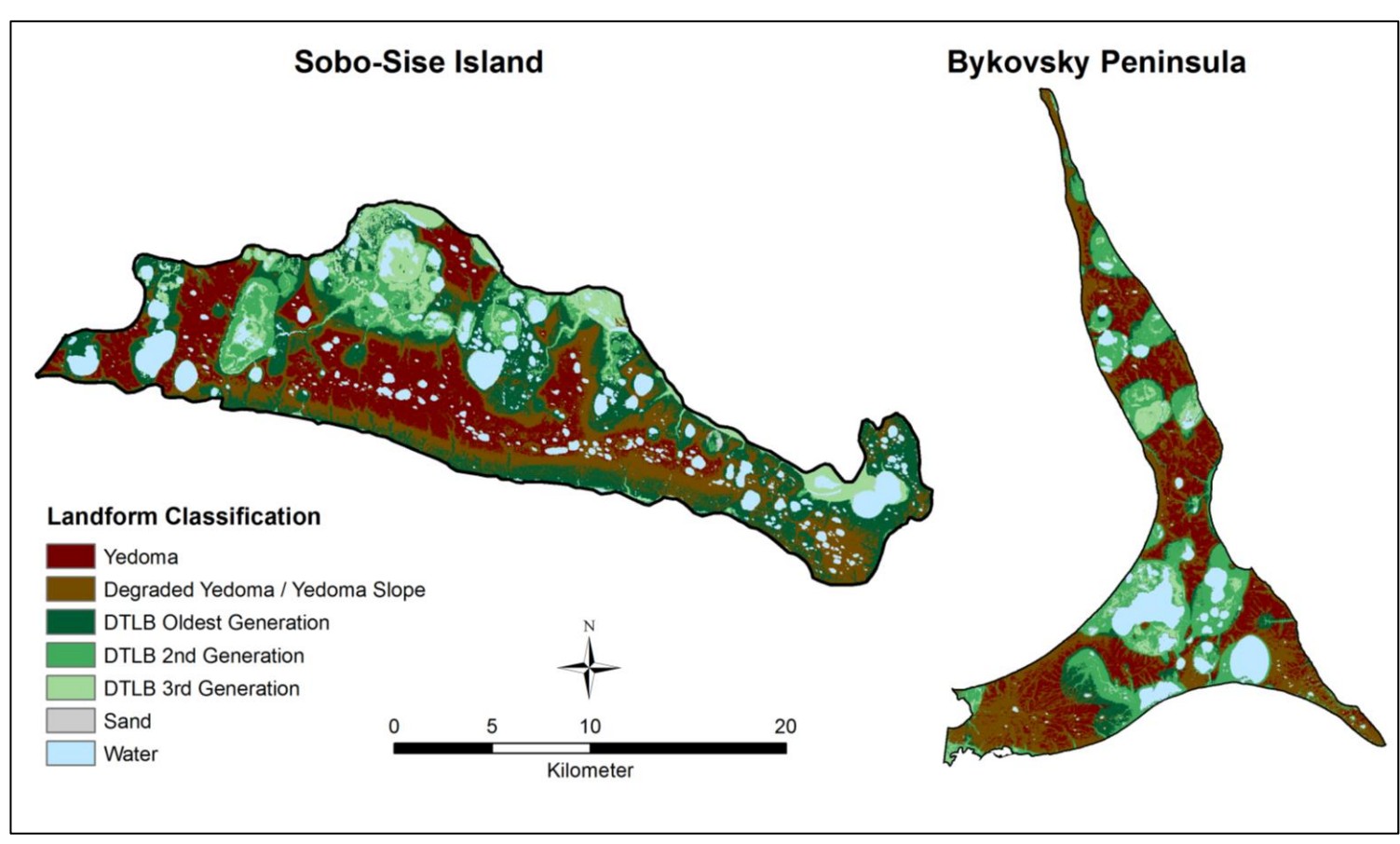

**Figure 5:** Landform classification of Sobo-Sise Island and Bykovsky Peninsula. For upscaling, the three classes of DTLB generations were merged into a single class 'Thermokarst'.

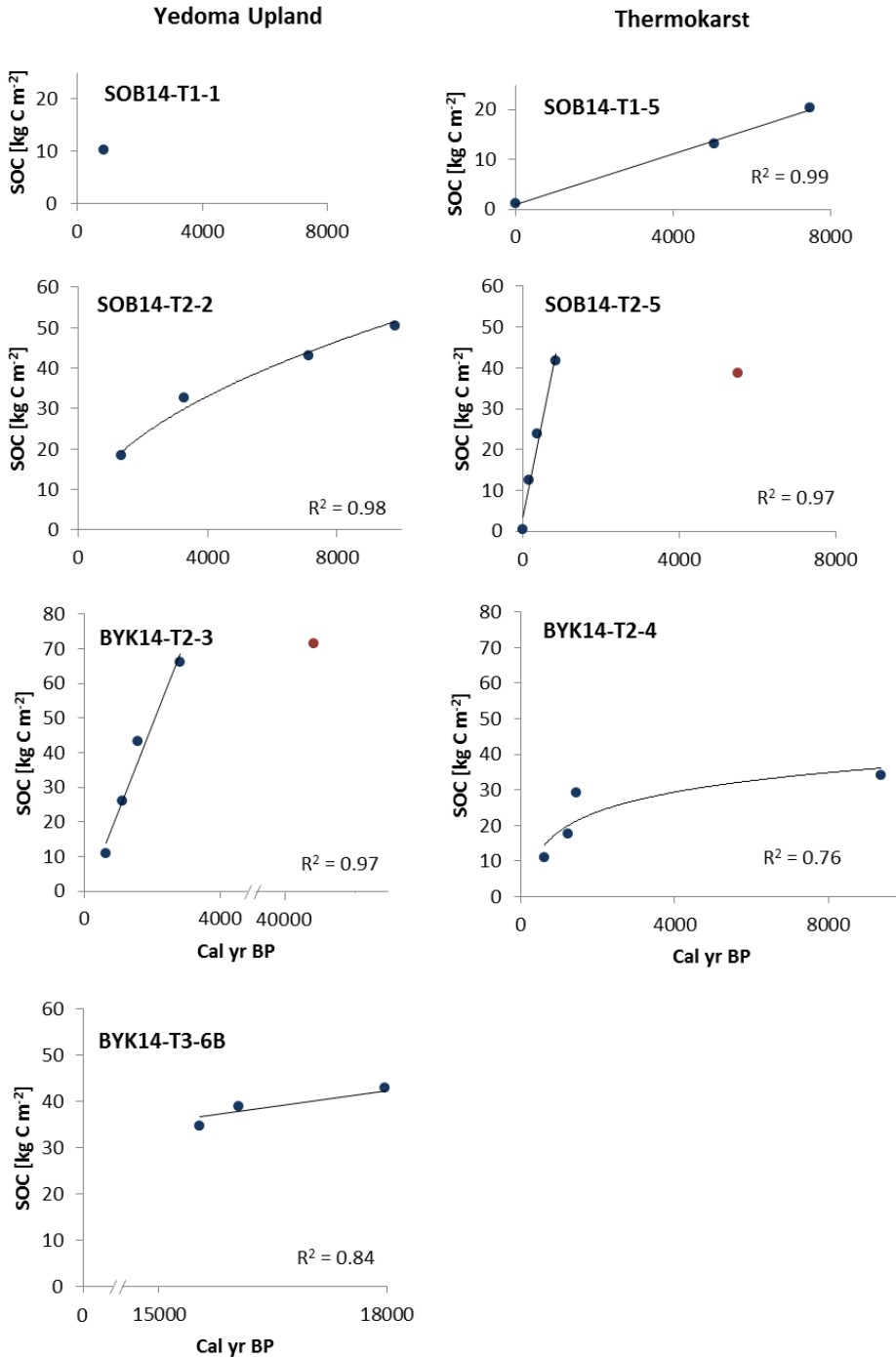

**Figure 6:** Relationship between cumulative SOC storage and age. Cal yr BP for radiocarbon dated samples (blue dots) for each core. Lines indicate the best fit correlation of the points excluding outliers (red dots).