# Peer review of "Carbon and nitrogen pools in thermokarst-affected permafrost landscapes in Arctic Siberia"

_Biogeosciences, 2017_

## Referee Comment (RC1) · Anonymous Referee #2 · 22 Oct 2017

The authors of "Carbon and nitrogen pools in thermokarst-affected permafrost land-scapes in Arctic Siberia" have executed and interesting study and presented a well written manuscript. The data are interesting and provide valuable measurements of permafrost C and N stocks. Some of the field sampling methods should be explained further to ensure the data merit scaling up using the landscape mapping approach de-scribed. The inclusion of N pools in this manuscript represent a novel contribution to the field of permafrost mapping but the implications for the author's findings should be more fully developed in the discussion. Details regarding the analysis of soil samples for %N must be clarified as well- the low N content of these mineral soils suggests that separate analysis for %C and %N would be necessary given the limitations of most

elemental analyzers.

Detailed comments and questions are listed below:

PG3, Line 11-15: Koven's study did an excellent job comparing two potentially limiting factors on ecosystems carbon balance (SOM-C decomposability, plant N limitation) but it doesn't present a strong argument against plants accessing N in newly thawed permafrost. They assume aboveground plant phenology reflects belowground plant phenology and say that they do not capture the microbial community/decomp dynamics observed in manipulative field experiments with the simplified N cycle they included in CLM. There is a growing body of field studies looking at plant access to N from thawing permafrost- Keuper et al 2017 (Global Change Biology) and references herein would be a good place to start.

PG3, Line 30: The introduction should include background information and literature references regarding the CN ratio of permafrost soils and how it relates to the past and future decomposition of SOM (ie, Schaedel et al 2014, Global Change Biology). The addition of N pool data is interesting component of this paper but seems underexplored.

PG5, Line 20-31: The field methods should be clarified to ensure the data represent the landscape adequately and will bear the scaling approaches utilized. How can the sampling point locations be equidistant from one another along a transect and reflect a stratified sample scheme? Does "stratified sampling" refers to the choice of transect location? How were the number of samples from baydzherakhs decide? What landscape features were baydzherakhs and DTLB data points grouped with for the average %C and %N values given in Table 1?

PG6, Line 9-13: Were separate samples run for %C and %N analysis? A larger weight of sample might have been necessary to determine %N numbers, especially on these mineral soils. A 5mg sample with only 0.1% N (Table 1 data) for instance would only have 5ug N which is likely below the detection limit for many elemental analyzers. Please provide detection range and sensitivity for this machine and/or specify sample

amounts for the separate analyses if indeed samples were run separately.

PG7, thoughout: Please clarify how ground-truthing was conducted. Were training areas selected based on observations made in the field? Did any of the field observations points overlap with the high-resolution imagery described to check landform classification?

PG8, Line 4-8: Somewhat confusing to have this discussion of ice wedge calculations when the reader does not yet know the source of ice wedge content data... consider moving to later in the methods section.

PG8, Line 27: Is the difference described here significant? Seems unlikely given the variation in the Bykovsky samples.

PG14, Line 12-17: Move this background info to Introduction. Would it be possible to use the C:N data in this paper to estimate C losses from these sites using the models in Schaedel et al. 2014? Some more developed discussion of potential N mineralization with decomposition of these soils would be warranted. This paper's inclusion of N stocks is novel but the discussion does not delve into the implications of the results.

Table 1: Please include symbols or alternate font styles to denote statistical differences between sites and landscape forms.

Figure 2: Model summaries for decreasing C:N with depth and summary statistics should be included here.

---

## Referee Comment (RC2) · Anonymous Referee #3 · 30 Oct 2017

Fuchs et al present analyses of soil carbon and nitrogen stocks across a range of representative landforms for two study areas in the Lena River Delta in northern Siberia. These areas are underlain by ice rich Pleistocene age yedoma permafrost, which is poorly represented in pan-Arctic inventories. The study quantifies variability within and between landforms, and uses high resolution multispectral and DEM data to create a landscape classification used for up-scaling soil carbon and nitrogen stocks. The methods of sampling and analyses are conventional, and executed well. The results are in line with other studies in that higher carbon and nitrogen stocks are found in yedoma soils. These results help improve understanding of landscape variability in permafrost soil properties. The nitrogen stocks are particularly useful, as these data are not often

analyzed or reported.

I do wonder if the data are open access, and if so, the authors should post a link to the repository or include a data citation with doi. This is becoming increasingly common, and this is a good thing. These data are valuable, especially given the remote location and importance of the data. The summaries in the supplement are a good start, but tabular data in a slightly more raw form would be good.

The manuscript will be suitable for publication pending a few relatively minor revisions.

Minor Comments: P3 L5: See also Webb et al 2017 – this is a recent citation and may be relevant here and in the discussion.

P3 L13: Perhaps cite Abbott et al 2016 here – I'm not sure that NPP increases will offset permafrost SOC emissions, even with increased N availability.

P3 L23: I don't think DTLB has been defined yet.

P9 L28-29: This statement is unclear – do you mean to say that a single core is affecting the mean? Please clarify.

P32-33: In figures 3 & 4 it would be good to have an outline around the key for the striped bars.

Abbott, B. W. et al. (2016), Biomass offsets little or none of permafrost carbon release from soils, streams, and wildfire: an expert assessment, Environ. Res. Lett., 11(3), 1–13, doi:10.1088/1748-9326/11/3/034014.

Webb, E. E. et al. (2017), Variability in above- and belowground carbon stocks in a Siberian larch watershed, Biogeosciences, 14(18), 4279–4294, doi:10.5194/bg-14-4279-2017.

---

## Author Comment (AC1) · 9 Dec 2017

Authors Reply by Matthias Fuchs, on behalf of all Co-authors (matthias.fuchs@awi.de)

We thank the Anonymous Referee for the valuable feedback which allows us to improve our manuscript. With this reply we hope to address all the comments and suggestion of the referee. Our changes and point-by-point replies are presented below.

Referee Comment (RC): The authors of "Carbon and nitrogen pools in thermokarst-affected permafrost Landscapes in Arctic Siberia" have executed and interesting study and presented a well written manuscript. The data are interesting and provide valuable

[Figure]

measurements of permafrost C and N stocks. Some of the field sampling methods should be explained further to ensure the data merit scaling up using the landscape mapping approach described.

Authors Reply (AR): Thank you for your positive feedback. We went through the field method sampling section and tried to improve it according to your suggestions. We hope to clarify with this reply the open questions.

RC: The inclusion of N pools in this manuscript represent a novel contribution to the field of permafrost mapping but the implications for the author's findings should be more fully developed in the discussion.

AR: This is a valuable suggestion and we included an additional paragraph in the discussion about the role of nitrogen in a thawing permafrost environment.

RC: Details regarding the analysis of soil samples for %N must be clarified as well- the low N content of these mineral soils suggests that separate analysis for %C and %N would be necessary given the limitations of most elemental analyzers.

AR: We did not run separate analyses for %C and %N. For the elemental analysis we homogenize and grinded the sub-samples thoroughly prior to measuring. Also, the elemental analyser does not measure the absolute weight of an element but through combustion and gas separation the peak flow of $N_2$ and $CO_2$ are detected and converted to the absolute weight of the element based on the calibration.

RC: PG3, Line 11-15: Koven's study did an excellent job comparing two potentially limiting factors on ecosystems carbon balance (SOM-C decomposability, plant N limitation) but it doesn't present a strong argument against plants accessing N in newly thawed permafrost. They assume aboveground plant phenology reflects belowground plant phenology and say that they do not capture the microbial community/decomp dynamics observed in manipulative field experiments with the simplified N cycle they included in CLM. There is a growing body of field studies looking at plant access to N

from thawing permafrost- Keuper et al 2017 (Global Change Biology) and references herein would be a good place to start.

AR: Thank you for the clarification. We removed Koven et al. (2015) from the introduction and added the suggested reference instead. We included a paragraph in the discussion about the potential offset of carbon release by the increased availability of nitrogen and increased biomass.

RC: PG3, Line 30: The introduction should include background information and literature references regarding the CN ratio of permafrost soils and how it relates to the past and future decomposition of SOM (ie, Schaedel et al 2014, Global Change Biology). The addition of N pool data is interesting component of this paper but seems underexplored.

AR: This is right and we agree that we did not cover the CN ratio in the introduction so far. We included now one additional paragraph in the introduction with background information regarding the CN ratio. In particular we moved PG14, Line 12-17 into the introduction as the referee suggested.

RC: PG5, Line 20-31: The field methods should be clarified to ensure the data represent the landscape adequately and will bear the scaling approaches utilized. How can the sampling point locations be equidistant from one another along a transect and reflect a stratified sample scheme? Does "stratified sampling" refers to the choice of transect location? How were the number of samples from baydzherakhs decide? What landscape features were baydzherakhs and DTLB data points grouped with for the average %C and %N values given in Table 1?

AR: The sampling approach is stratified in the sense, that we chose the start location of the transect, the direction and the distance between the points. Randomness is included by the fixed distance between the sample points. The decision of the starting point and which landform types we want to cover with our transect is therefore stratified. Baydzherakhs are a common but rather small feature in the study area. Due to

the transect approach and the points equidistant from each other, we did not end up on a baydzherakh in two transects. On one of those transects (SOB14-T1) there were no baydzherakhs in the surroundings, on the other transect (SOB14-T2) we additionally sampled a baydzherakh in close proximity to the transect. On another transect on Bykovsky Peninsula (BYK14-T2) we chose a baydzherakh as a starting point. Baydzerakhs represent polygon centres; since we want to drill permafrost soil cores it is imperative to drill in polygon centres and baydzerakhs a perfect example for this. With the transect approach, we sampled different Yedoma upland sites, however due to an ice wedge content of 40 – 44 % in the landscape, we sometimes hit ice-wedges while sampling. The baydzheraks are seen as additional sampling sites on transects where they occurred. However, even for a RapidEye based upscaling, these features are too small to be resolved in a landform classification. Baydzherakhs were grouped into the Yedoma upland class and DTLB data points were grouped into the thermokarst class in Table 1.

RC: PG6, Line 9-13: Were separate samples run for %C and %N analysis?

AR: The TC and TN measurements are measured in one run. Just TOC is measured separately with another device. However, each sample was measured with two replicates and we allow a deviation < 5% between the two measurements. If this criterion is not fulfilled, we repeat the measurements until our data quality criterion is met. All lab data is checked by the laboratory leads before the data is released.

RC: A larger weight of sample might have been necessary to determine %N numbers, especially on these mineral soils. A 5mg sample with only 0.1% N (Table 1 data) for instance would only have 5ug N which is likely below the detection limit for many elemental analyzers. Please provide detection range and sensitivity for this machine and/or specify sample amounts for the separate analyses if indeed samples were run separately.

AR: A larger weight of the samples might increase the accuracy of the measurements

in case of very low percentages. However, the elemental analyser does not measure the absolute weight or percentages but it combusts the sample and measures the peak flow of N2 and CO2 (in mV) and then calculates the %C and %N based on the calibration, the integral of the peak flow and the initial weight of the sample. Also, we annually calibrate the device in accordance with Elementar Analysesysteme (the manufacturer of the device). This calibration determines the maximum and minimum threshold for the measurements and therefore determines the minimum detection limit which is approximately three times above the measured noise (personal communication with Elementar Analysesysteme). We start each measurement run with 8 standards (EDTA, IVA) for determining the daily factors of C and N. This is followed by another 8 control standard samples (EDTA, IVA, soil standards). This control group is always included after 30 measured samples as well as in the end of each measurement run. With this procedure we can check the accuracy and detect potential imprecisions during the measurement. Prior to the measurement we homogenize the samples by grinding thoroughly. We measure two replicates of each sample and allow a <5% deviation for our double measurement. The device gives percentage values if the measurement is in the detection range. Otherwise an error is reported. The sensitivity of the Vario EL III is <0.1%.

RC: PG7, thoughout: Please clarify how ground-truthing was conducted. Were training areas selected based on observations made in the field? Did any of the field observations points overlap with the high-resolution imagery described to check landform classification?

AR: The high resolution images do overlap with the sites sampled in the field. However, field time, as usually, is limited and our field-based ground truth sites had to be complemented by ground truth points characterize with very high-resolution imagery, with which it was possible to differentiate between the different landform types as well as water areas. Such mixed ground truth approaches have been successfully applied before in areas with limited field data. The training areas for the land cover classification were chosen based on the field reconnaissance (PG7 Line 22). Further a digital elevation model helped to identify partly degraded Yedoma uplands.

RC: PG8, Line 4-8: Somewhat confusing to have this discussion of ice wedge calculations when the reader does not yet know the source of ice wedge content data... consider moving to later in the methods section.

AR: We removed the ice wedge calculation part from this paragraph and mentioned it in the subsequent paragraph together with the ice wedge content data.

RC: PG8, Line 27: Is the difference described here significant? Seems unlikely given the variation in the Bykovsky samples.

AR: The difference is statistically significant (student t-test; $p < 0.01$). However, in an earlier review phase, we decided to keep the manuscript descriptive.

RC: PG14, Line 12-17: Move this background info to Introduction.

AR: Changed as suggested. See comment above on PG 3 line 30.

RC: Would it be possible to use the C:N data in this paper to estimate C losses from these sites using the models in Schaedel et al. 2014? Some more developed discussion of potential N mineralization with decomposition of these soils would be warranted. This paper's inclusion of N stocks is novel but the discussion does not delve into the implications of the results.

AR: According to Schädel et al. (2014), the % initial C in combination with the CN ratio can be an indicator for the potential C loss over time. However, the scope of the paper was not the determination of C release but the estimate of C pools across different depth intervals in a thermokarst landscape. Hence, detailed modelling would go to far for this study but could be envisioned in the future. We added a paragraph on nitrogen which becomes available upon permafrost thawing and its potential effects on biomass production and carbon release offset in the discussion chapter 4.1.

RC: Table 1: Please include symbols or alternate font styles to denote statistical differences between sites and landscape forms.

AR: In a previous review phase we decided to keep the manuscript descriptive, since this was suggested by the editor.

RC: Figure 2: Model summaries for decreasing C:N with depth and summary statistics should be included here.

AR: The correlations of a decreasing CN with depth are added to the graph. The mean CN for active and permafrost layer are mentioned in the text on page 9, line 12-15.

References:

Keuper, F. et al. (2017): Experimentally increased nutrient availability at the permafrost thaw front selectively enhances biomass production of deep-rooting subarctic peatland species, Glob. Change. Biol., 23, 4257-4266, doi:10.1111/gcb.13804.

Schädel, C., et al. (2014): Circumpolar assessment of permafrost C quality and its vulnerability over time using long-term incubation data, Glob. Change Biol., 20, 641-652, doi:10.1111/gcb.12417.

We thank the Anonymous Referee for the constructive comments which allow us to improve our manuscript.

On behalf of all the Co-authors,

Matthias Fuchs

---

## Author Comment (AC2) · 9 Dec 2017

Authors reply Matthias Fuchs, on behalf of all Co-authors (matthias.fuchs@awi.de)

We thank the Anonymous Referee for the positive feedback on our manuscript and the valuable comments. We have gone through your comments and improved our manuscript according to your suggestions. Also, we uploaded the data in the PANGAEA data repository and will provide a doi link in the final version of our manuscript. We hope to address with this reply all your suggestions and questions. Our changes and point-by-point replies are presented below.

[Figure]

Referee Comment (RC): Fuchs et al present analyses of soil carbon and nitrogen stocks across a range of representative landforms for two study areas in the Lena River Delta in northern Siberia. These areas are underlain by ice rich Pleistocene age yedoma permafrost, which is poorly represented in pan-Arctic inventories. The study quantifies variability within and between landforms, and uses high resolution multispectral and DEM data to create a landscape classification used for up-scaling soil carbon and nitrogen stocks. The methods of sampling and analyses are conventional, and executed well. The results are in line with other studies in that higher carbon and nitrogen stocks are found in yedoma soils. These results help improve understanding of landscape variability in permafrost soil properties. The nitrogen stocks are particularly useful, as these data are not often analyzed or reported.

Authors Reply (AR): We thank the referee for this positive feedback and acknowledging the importance of our study.

RC: I do wonder if the data are open access, and if so, the authors should post a link to the repository or include a data citation with doi. This is becoming increasingly common, and this is a good thing. These data are valuable, especially given the remote location and importance of the data. The summaries in the supplement are a good start, but tabular data in a slightly more raw form would be good.

AR: We submitted the data to the PANGAEA data repository (www.pangaea.de). With the following link the data is accessible:

https://doi.pangaea.de/10.1594/PANGAEA.883582

We submitted the data from Table 1 from the manuscript, the sample site characteristics, the carbon and nitrogen data presented in the supplement material, as well as the radiocarbon data to the PANGAEA data repository. We will include the doi link in the final version of our paper.

RC: P3 L5: See also Webb et al 2017 – this is a recent citation and may be relevant

here and in the discussion.

AR: Thank you for the suggestion. We included this citation in the introduction and added a sentence in the discussion.

RC: P3 L13: Perhaps cite Abbott et al 2016 here – I'm not sure that NPP increases will offset permafrost SOC emissions, even with increased N availability.

AR: We changed this part of the introduction slightly according to referee #2. Though, we included the suggested reference in the discussion.

RC: P3 L23: I don't think DTLB has been defined yet.

AR: Yes you are right, thank you for the hint. We included the definition of this abbreviation in the sentence on P3 L20-24.

RC: P9 L28-29: This statement is unclear – do you mean to say that a single core is affecting the mean? Please clarify.

AR: For thermokarst on Bykovsky Peninsula we only have one core (BYK14-T3-3) reaching a depth of two meters. Therefore, the mean for 0-200 cm is based on this single core only (see Fig. 3). This core was very carbon rich compared to the other cores (see Table S4) from thermokarst sites and therefore the high carbon storage for thermokarst in 0-200 cm need to be interpreted carefully. We added the following sentence in the manuscript on P10 L7: "However, this estimation is only based on one relatively C rich core (BYK14-T3-3), since this is the only core reaching a depth of two meters for thermokarst on Bykovsky Peninsula. Therefore the carbon estimation for thermokarst on Bykovsky Peninsula for the soil interval 0-200 cm has to be interpreted carefully."

RC: P32-33: In figures 3 & 4 it would be good to have an outline around the key for the striped bars.

AR: Changed as suggested

References:

Abbott, B. W. et al. (2016), Biomass offsets little or none of permafrost carbon release from soils, streams, and wildfire: an expert assessment, Environ. Res. Lett., 11(3), 1–13, doi:10.1088/1748-9326/11/3/034014. Webb, E. E. et al. (2017), Variability in above- and belowground carbon stocks in a Siberian larch watershed, Biogeosciences, 14(18), 4279–4294, doi:10.5194/bg-14-4279-2017.

We thank the Anonymous Referee for the constructive comments and suggestions.

On behalf of all the Co-authors

Matthias Fuchs